# Profiling neuronal methylome and hydroxymethylome of opioid use disorder in the human orbitofrontal cortex

Gregory Rompala[1,15], Sheila T. Nagamatsu [2,3,4,15],
José Jaime Martínez-Magaña [2,3,4], Diana L. Nuñez-Ríos[2,3,4], Jiawei Wang [5,6],
Matthew J. Girgenti [2,4], John H. Krystal [2,3,4], Joel Gelernter [2,3,4],
Traumatic Stress Brain Research Group*, Yasmin L. Hurd [1] &
Janitza L. Montalvo-Ortiz [2,3,4] ✉

Opioid use disorder (OUD) is influenced by genetic and environmental factors. While recent research suggests epigenetic disturbances in OUD, this is mostly limited to DNA methylation (5mC). DNA hydroxymethylation (5hmC) has been widely understudied. We conducted a multi-omics profiling of OUD in a male cohort, integrating neuronal-specific 5mC and 5hmC as well as gene expression profiles from human postmortem orbitofrontal cortex (OUD = 12; non-OUD = 26). Single locus methylomic analysis and co-methylation analysis showed a higher number of OUD-associated genes and gene networks for 5hmC compared to 5mC; these were enriched for GPCR, Wnt, neurogenesis, and opioid signaling. 5hmC marks also showed a higher correlation with gene expression patterns and enriched for GWAS of psychiatric traits. Drug interaction analysis revealed interactions with opioid-related drugs, some used as OUD treatments. Our multi-omics findings suggest an important role of 5hmC and reveal loci epigenetically dysregulated in OFC neurons of individuals with OUD.

Opioid use disorder (OUD) is a serious public health problem because of its high disease burden, as measured by hospitalization and drug overdose death rates[1], and involvement in the criminal justice system based on high incarceration rates[2]. In the U.S., opioid overdose deaths reach 17.8 per 100,000 individuals. This opioid epidemic worsened during the COVID-19 pandemic, as shown by a significant steep rise of 29.4% in opioid overdose deaths in 2020[3].

OUD is associated with a wide range of acute effects and long-term brain neuroadaptations related to intoxication, tolerance, and dependence, which can contribute to compulsive opioid use[4,5]. Activation of μ, ∂, and k opioid receptors alters the activity of stress and reward circuitry. In both animals and humans, the orbitofrontal cortex (OFC) has been implicated in the development and maintenance of drug addiction, deficits in the inhibition of impulsive behavior, and distortions in reward-related decision-making processes[6–9].

Multiple epigenetic modifications regulate gene expression that impacts behaviors relevant to opioid addiction[10]. We and others have evaluated the relationship between 5mC and opioid-related traits in human peripheral tissue and postmortem brain tissue. For example, an epigenome-wide association study (GWAS) in European women identified three differential DNA methylated sites associated with opioid dependence in peripheral blood[11]. In the human postmortem brain, a

[1]Icahn School of Medicine at Mount Sinai, New York, NY, USA. [2]Department of Psychiatry, Yale University School of Medicine, New Haven, CT, USA. [3]VA Connecticut Healthcare System, West Haven, CT, USA. [4]U.S. Department of Veterans Affairs National Center for Posttraumatic Stress Disorder, Clinical Neurosciences Division, West Haven, CT, USA. [5]Computational Biology and Bioinformatics Program, Yale University, New Haven, CT, USA. [6]Department of Biostatistics, Yale School of Public Health, New Haven, CT, USA. [15]These authors contributed equally: Gregory Rompala, Sheila T. Nagamatsu. *A list of authors and their affiliations appears at the end of the paper. ✉e-mail: janitza.montalvo-ortiz@yale.edu

study evaluating 5mC in individuals that experienced heroin abuse who died from overdose reported differential 5mC of several gene classes, including those implicated in glutamate neurotransmission, axonogenesis, synaptic processes, and the regulation of gene expression[12]. A more recent 5mC study in the dorsolateral prefrontal cortex (dlFPC) reported 13 CpG sites nominally associated with opioid intoxication ($p < 1.0 \times 10^{-5}$); with no CpG sites surviving multiple testing correction[13]. Another recent study in the dlFPC conducted an integrative analysis of epigenomic and transcriptomic data in the context of OUD, identifying potential regulatory genes associated with OUD-related expression patterns, and co-methylated modules associated with OUD-enriched neurogenesis, nervous system development, and generation of neurons[14].

5mC is actively oxidized to hydroxymethyl-cytosine (5hmC) by ten-eleven translocation (TET) enzymes. Recent work has found that 5hmC is a relatively stable epigenetic mark, highly enriched in neurons, and associated with transcription activation[15]. Studies have also shown that 5hmC plays a key role in brain development processes[16]. Genome-wide differential 5hmC studies in the human postmortem brain have suggested an association with antemortem psychiatric traits, including depression[17] and alcohol use disorders[18]. Although there are no prior studies of 5hmC in human OUD, a study in rodents chronically exposed to morphine reported changes in both global and promoter-specific 5mC and 5hmC levels across multiple brain regions[19].

To date, epigenetic studies in postmortem human brain tissue have been limited to analyses of bulk tissue samples containing diverse neuronal and non-neuronal cell types. Since epigenetic marks regulate gene expression in a cell type-specific manner, the presence of mixed cell types in bulk tissue may obscure cell type-specific findings.

In this study, we conducted a parallel 5mC and 5hmC profiling of OUD in neuronal nuclei from the human postmortem OFC of a male cohort. Further, we integrated gene expression data to elucidate the impact of differential 5mC and 5hmC on gene transcription. Lastly, we performed a GWAS enrichment analysis to investigate whether differential 5mC and 5hmC marks are associated with genetic variations linked to OUD. Our integrative multi-omics study uncovered functional marks of neuronal 5mC and 5hmC, as well as co-methylation networks associated with OUD in the human OFC.

## Results

### Demographics and clinical characteristics

Brain samples from the OFC included 12 OUD cases and 26 individuals without OUD (non-OUD group) (Table 1). All brain samples were from males of European and African ancestry. There was no significant

**Table 1 | Demographics summary**

|  | OUD− (N = 26) | OUD+ (N = 12) | P value |
|---|---|---|---|
| Age (μ ± SD) | 43.1 ± 11.55 | 37.6 ± 8.9 | 0.1518 |
| PMI (μ ± SD) | 30.65 ± 8.15 | 29.6 ± 7.5 | 0.7200 |
| Ancestry |  |  |  |
| AA | 5 | 4 | 0.4428 |
| EA | 19 | 8 |  |
| Cigarette smoking | 13 | 10 | 0.0770 |
| Alcohol dependence | 5 | 3 | 0.6893 |
| PTSD | 13 | 12 | 0.0026 |
| Major depressive disorder | 5 | 8 | 0.0086 |
| Polysubstance abuse | 4 | 7 | 0.0174 |
| Removed outliers | 2 | 0 | 1.0000 |

The table shows the main information for individuals without OUD (OUD−) and with OUD (OUD+). A statistical test was performed to evaluate if the OUD− group differed from OUD+ group. The *P* value indicated was calculated using the Fisher test.
*AA* African American, *EA* European American.

difference in the age of death between the OUD and non-OUD groups ($p$ value = 0.15), or in smoking status ($p$ value = 0.0770). However, we did observe significant differences in the prevalence of post-traumatic stress disorder (PTSD; $p$ value = 0.0026), and major depressive disorder (MDD) ($p$ value = 0.0086).

### Assessment of 5mC and 5hmC in OFC Neurons

We utilized fluorescence-activated nuclei sorting (FANS) with the postmitotic neuronal marker NeuN and reduced representation oxidative bisulfite sequencing (RRoxBS) to profile 5mC and 5hmC in OFC neuronal nuclei isolated from postmortem brain samples (Fig. 1a, Supplementary Fig. 1). All CpG sites (CpGs) with ≥10 base pair coverage was analyzed, with most CpGs having coverage between 40 and 60× (Fig. 1b). An average of 10× coverage was obtained for ~3.5 million CpGs. Overall, we analyzed 1,844,968 CpGs for 5mC and 1,653,870 for 5hmC. Principal component analysis of all 5mC and 5hmC samples identified two outlying subjects, which were subsequently removed from the final analysis (Supplementary Fig. 2). The majority of CpGs (44%) were in gene promoter regions (±1 kb from the transcription start sites−TSS), followed by intergenic regions (30%), intronic regions (22%), and exonic regions (4%) (Fig. 1c). We observed a significant reduction in 5mC and 5hmC levels in the promoter and intron regions of neuronal-marker genes compared to non-neurons markers, although 5hmC was increased in exons for neuronal genes (Fig. 1d, e). Furthermore, we analyzed previously established differentially methylated regions in cortical neuronal nuclei[20] and found a strong relationship between total methylation (5mC+5hmC) at CpGs in neurons of that dataset and neuronal OFC total methylation of the same CpGs in this study ($R^2 = 0.66$, $p < 0.0001$; Fig. 1f). Similarly, CpG methylation in OFC NeuN+ neurons showed an inverse correlation with CpG methylation previously observed in cortical non-neuronal nuclei ($R^2 = -0.44$, $p < 0.0001$; Fig. 1f). We observed reduced 5mC at neuronal-marker genes compared to non-neuronal genes, which is consistent with our sample being neuronal-specific. We anticipate a permissive 5mC signature promoting active gene expression[15] and a more repressive 5mC signature at non-neuronal genes. Moreover, in line with 5hmC being associated with active gene expression in the gene body (mainly exons), we demonstrate that 5hmC levels are higher in neuronal-specific (active) genes compared to non-neuronal-specific (inactive) genes.

### Differential 5mC and 5hmC associated with OUD

Individual CpGs were evaluated for differential 5mC and 5hmC levels in individuals with OUD compared to those without OUD (top genes indicated in Table 2). For 5mC, we found 397 differential CpGs (357 genes) (Supplementary Data 1); while for 5hmC, we identified 1740 differential CpGs (1453 genes) (Supplementary Data 2). The $\lambda$ values for 5mC and 5hmC were 1.07 and 0.92, respectively. QQ plots are included in Supplementary Fig. 3. There was no overlap between the differential 5mC and 5hmC marks. However, at the gene level, we observed 38 overlapping genes between the differential 5mC and 5hmC marks (9.6% of 5mC-linked genes and 2.2% of 5hmC-linked genes) (Fig. 2a, b). Evaluating differential CpGs across genomic loci indicated that most 5mC and 5hmC CpGs were in promoter regions within 1 kb of a TSS. (Fig. 2c). Moreover, examining genomic regions with known associations with the histone modification H3K27ac indicates that these CpGs likely reside in active promoter and enhancer regions[21]. Furthermore, the enrichment was more pronounced in neuron-specific regions compared to non-neuronal-specific H3K27ac regions (Fig. 2d, Supplementary Data 3). Differential 5mC marks were most significantly enriched in GABAergic neuronal-marker genes, while differentially 5hmC CpGs were significantly enriched in neuronal-marker genes (i.e., genes increased in both GABAergic and glutamatergic neurons compared to non-neuronal cell types) (Fig. 2e, Supplementary Data 3). The differences observed between the differential 5mC and 5hmC CpGs, in terms

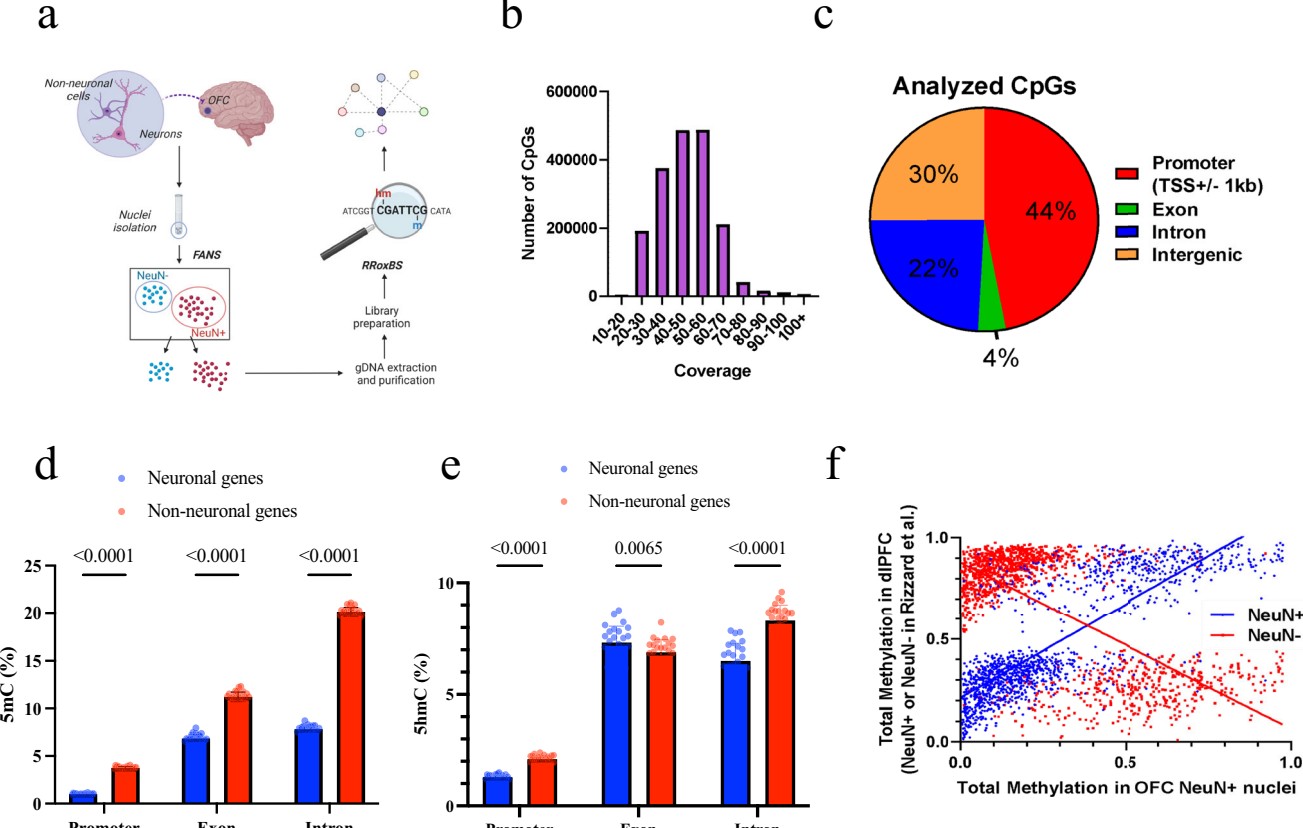

**Fig. 1 | Analyzing the neuronal methylome and hydroxymethylome in OUD. a** Experimental workflow. Fluorescence-activated nuclei sorting was used to isolate neuronal nuclei from postmortem OFC. Nuclei were processed for genomic DNA, undergoing reduced representation oxidative bisulfite sequencing to examine 5mC and 5hmC at CpG-dense loci. Figure created with BioRender.com. **b** An average of 10x coverage was obtained for ~3.5 million CpG sites. **c** 44% of CpG sites were in promoter regions and 30% in intergenic regions. **d**, **e** Neuronal 5mC occurs mainly in intergenic regions, while 5hmC occurs in introns, exons, and intergenic regions (n = 38 biologically independent samples; 26 OUD+; 12 OUD−). Data are presented as mean values ± SD, and p value were generated using ANOVA. **f** Contrasting mean 5mC levels at neuronal (NeuN+) vs. non-neuronal (NeuN−) marker genes.

of cell type enrichment and CpG location, suggest distinct gene regulatory mechanisms among these two epigenetic modifications. Gene ontology (GO) enrichment analyses revealed no significant pathways among the OUD-associated 5mC genes. However, trending top ontologies (FDR-adjusted p value < 0.1), included nicotine- and opioid-signaling pathways (Fig. 2g, Supplementary Data 4). In comparison, OUD-associated 5hmC genes were significantly enriched for several terms (FDR-adjusted p value < 0.05), many of which were related to neuronal function (e.g., G protein signaling, postsynaptic differentiation, and GABAB receptor signaling). Opioid-signaling pathways were also observed (FDR-adjusted p value = 0.05, Fig. 2h, Supplementary Data 5).

We compared our findings with those of Kozlenkov et al.[12], who examined 5mC in the postmortem OFC of individuals with a history of heroin use that died from opioid overdose (N = 87) using the Illumina 450 K Infinium microarray. Very few differentially methylated CpGs identified in their study had sufficient read coverage to be examined in the present study (175 of 1391). None of these sites showed differential 5mC or 5hmC in our study. This discrepancy may be due to technical differences, low coverage in those CpG sites, or demographic differences between the studies. However, when comparing genes with differential CpGs and using the same significance threshold as in Kozlenkov et al.[12] (p 0.001), we found that 256 CpGs were commonly differentially methylated between opioid overdose and OUD. The overlap increased to 327 CpGs when considering both 5mC and 5hmC. (Fig. 2f, Supplementary Data 6). GO analysis revealed that the overlapped genes with Kozlenkov et al. were associated with axon guidance

(5mC + 5hmC; odds ratio = 3.08, FDR-adjusted p value < 0.001, Supplementary Data 7).

We also assessed the overlap between our findings and the existing literature on opioid epigenomics. When considering recent 5mC studies evaluating the dorsolateral prefrontal cortex in relation to opioid-related traits, we identified an overlap of three genes (SMARCA4;[14] JUN;[14] TAF3[13]). In terms of 5mC studies in peripheral tissues, we found one overlapping gene from an EWAS study of opioid self-administration in patients who underwent dental surgery using saliva tissue[22]. Furthermore, we identified 127 overlapping genes when compared to two studies on the placenta investigating neonatal opioid withdrawal syndrome[23,24], and six overlapping genes in a study examining recent opioid medication use[25] (Supplementary Data 8).

To ascertain the potential impact of comorbid post-traumatic stress disorder on our findings, we examined the gene-level overlap between our results and previously 5mC associations with PTSD ("PTSD-associated genes" in Supplementary Data 1 and 2). We found minimal overlap when considering both 5mC and 5hmC differential sites, indicating that our findings are not primarily driven by comorbid PTSD effects.

**Gene-drug interaction analyses of differential 5mC and 5hmC**

We assessed predicted drug interactions for the annotated genes of the differential 5mC and 5hmC CpGs and identified 616 interactions for 5mC and 2,562 interactions for 5hmC (Supplementary Data 9 and 10). Additionally, we observed interactions with 15 opiates (Fig. 3). Among the differential 5mC CpGs, we found a single interaction with

**Table 2 | Top 10 genome-wide significant OUD-associated CpGs from the 5mC and 5hmC differential analysis**

| 5mC | | | | | |
|---|---|---|---|---|---|
| Chr | Start | q value | Δ 5mC | Nearest gene ID | Position to the nearest gene ID |
| chr19 | 54096075 | 1.13E-289 | −0.40 | OSCAR | Gene_body |
| chr7 | 108569763 | 4.00E-240 | 0.07 | THAP5 | Upstream |
| chr11 | 128891189 | 3.19E-185 | −0.21 | KCNJ5 | Downstream |
| chr14 | 99480992 | 4.06E-183 | 0.22 | SETD3 | Upstream |
| chr19 | 56120943 | 2.36E-121 | −0.21 | ZNF787 | Gene_body |
| chr14 | 105526298 | 8.33E-86 | 0.12 | TMEM121 | Downstream |
| chr11 | 3165259 | 4.65E-66 | 0.52 | OSBPL5 | Gene_body |
| chr6 | 12008931 | 1.95E-54 | −0.26 | HIVEP1 | Gene_body |
| chrX | 9465553 | 6.36E-53 | −0.19 | TBL1X | Gene_body |
| chr9 | 126805077 | 6.36E-53 | −0.37 | ZBTB43 | Gene_body |
| 5hmC | | | | | |
| Chr | Start | q value | Δ 5hmC | Gene symbol | Position to the nearest gene ID |
| chr1 | 109042116 | 1.05E-306 | −0.22 | WDR47 | Upstream |
| chr16 | 67416523 | 2.02E-306 | −0.15 | ZDHHC1 | Gene_body |
| chr6 | 163413930 | 2.72E-305 | 0.03 | CAHM | Gene_body |
| chr17 | 27793835 | 3.12E-291 | −0.21 | NOS2 | Gene_body |
| chr5 | 181190518 | 3.74E-291 | −0.21 | LINC01962 | Gene_body |
| chr19 | 639763 | 6.76E-291 | 0.40 | FGF22 | Downstream |
| chr4 | 37891265 | 2.85E-290 | −0.31 | TBC1D1 | Gene_body |
| chr9 | 120928860 | 5.78E-287 | 0.08 | TRAF1 | Gene_body |
| chr5 | 149960747 | 6.32E-260 | −0.22 | SLC26A2 | Downstream |
| chr2 | 96760842 | 6.92E-260 | −0.19 | CNNM4 | Downstream |

The table includes the chromosomal location, q value statistics, effect size, annotated gene using the nearest Gene ID, and position to the nearest gene ID (i.e., downstream, upstream, or in the gene body, considering a maximum TSS distance of 1500 bp).

apomorphine in the *CALY* gene (with the 5mC CpG located in the gene body). In the case of 5hmC, we found predicted interactions between opioids with seven genes: *CBFB, GRIN1, HCN1, HMOX2, MPO, RUNX1,* and *SOD2. CBFB, HCN1, HMOX2, MPO, RUNX1,* and *SOD2*. The CpGs in those genes were in the gene body, while the CpG annotated in the *GRIN1* was downstream (<200 bp) of the gene. Among all the genes with differential 5hmC, only *HCN1* interacted exclusively with opioids (specifically tramadol). In addition to opioids, *MPO* interacts with other drugs, including anti-inflammatory drugs (i.e., nimesulide, tolmetin, and diclofenac). *GRIN1* also showed interactions with glutamate receptor drugs (i.e., dizocilpine, (d)-serine), and pain management drugs (e.g., ketamine, ralfinamide, orphenadrine, orphenadrine citrate). *CBFB* interacts with a dopamine receptor agonist drug called ergocornine.

**Co-methylation analysis of OUD**

In addition to conducting EWAS, we performed co-methylation analysis using weighted gene co-expression network analysis (WGCNA) to identify clusters of CpG sites (modules) with similar methylation levels. This approach allowed us to assess the inter-correlation among CpG sites and identify epigenetically defined gene networks associated with OUD. A similar co-methylation analysis has been previously applied in the context of alcohol use disorder in the human postmortem brain[26]. Our co-methylation analysis identified 626 modules for 5mC and 572 for 5hmC. We analyzed module eigengene associated with OUD (|cor|> ±0.4 and *p* value < 0.05), resulting in 10 modules for 5mC (Fig. 4a) and four modules for 5hmC (Fig. 4b). Module membership versus gene significance is shown in Supplementary Fig. 5. Among the 5mC modules, six showed enrichments for GO terms (Supplementary Data 11), with the top 10 terms displayed in Fig. 4c. The 5mC module that exhibited the strongest correlation with OUD (cor = 0.48) was the Steelblue1 module,

enriched for Pre-NOTCH Transcription and Translation (*p* value = 5.71E-03; Supplementary Data 12. Most of the OUD-associated 5mC modules showed enrichment for transcription regulation, cell differentiation, nervous system development, morphogenesis, and generation of neurons. Additionally, two of these modules, turquoise3, and lightpink2, displayed enrichment for Reactome pathways, including inhibition of Voltage-Gated Ca²⁺ Channels via Gbeta/gamma Subunits, activation of GABAB Receptors (*p* value range: 1.44E-02 to 6.55E-03, turquoise3), angiotensin II-stimulated signaling through G proteins and beta-arrestin and Signaling by WNT (*p* value range: 1.38E-02 to 4.44E-02, lightpink2). The OUD-associated 5hmC modules showed enrichment for organ development, nervous system development, neurogenesis, transcription regulation, and morphogenesis (Supplementary Data 13, Fig. 4d). Enriched Reactome pathways (Supplementary Data 12) included GPCR Ligand Binding (*p* value = 1.77E-03; thistle1), Class B/2 (Secretin Family Receptors) (*p* value = 7.60E-03, rosybrown1), PI3K Cascade, and Insulin Receptor Signaling Cascade (*p* value range: 1.18E-02 to 4.95E-02, orange4). In our module-based analysis, we found *OPRM1* in the lightpink2 module, a gene commonly reported in the opioid literature.

Protein-protein interaction (PPI) analyses were performed for the OUD-associated 5mC and 5hmC modules (Fig. 5) using co-expression evidence. The PPI networks showed enrichment for several biological pathways, including regulation of biological process (GO:0050789), nitrogen compound metabolic process (GO:0006807), neurogenesis (GO:0022008), cell differentiation (GO:0030154), gene expression (GO:0010467), neuron development (GO:0048666), regulation of neurogenesis (GO:0050767), regulation of cell communication (GO:0010646), and response to stimulus (GO:0050896).

We also assessed the enrichment of differential 5mC and 5hmC marks in the identified OUD-associated module networks. For the 5mC

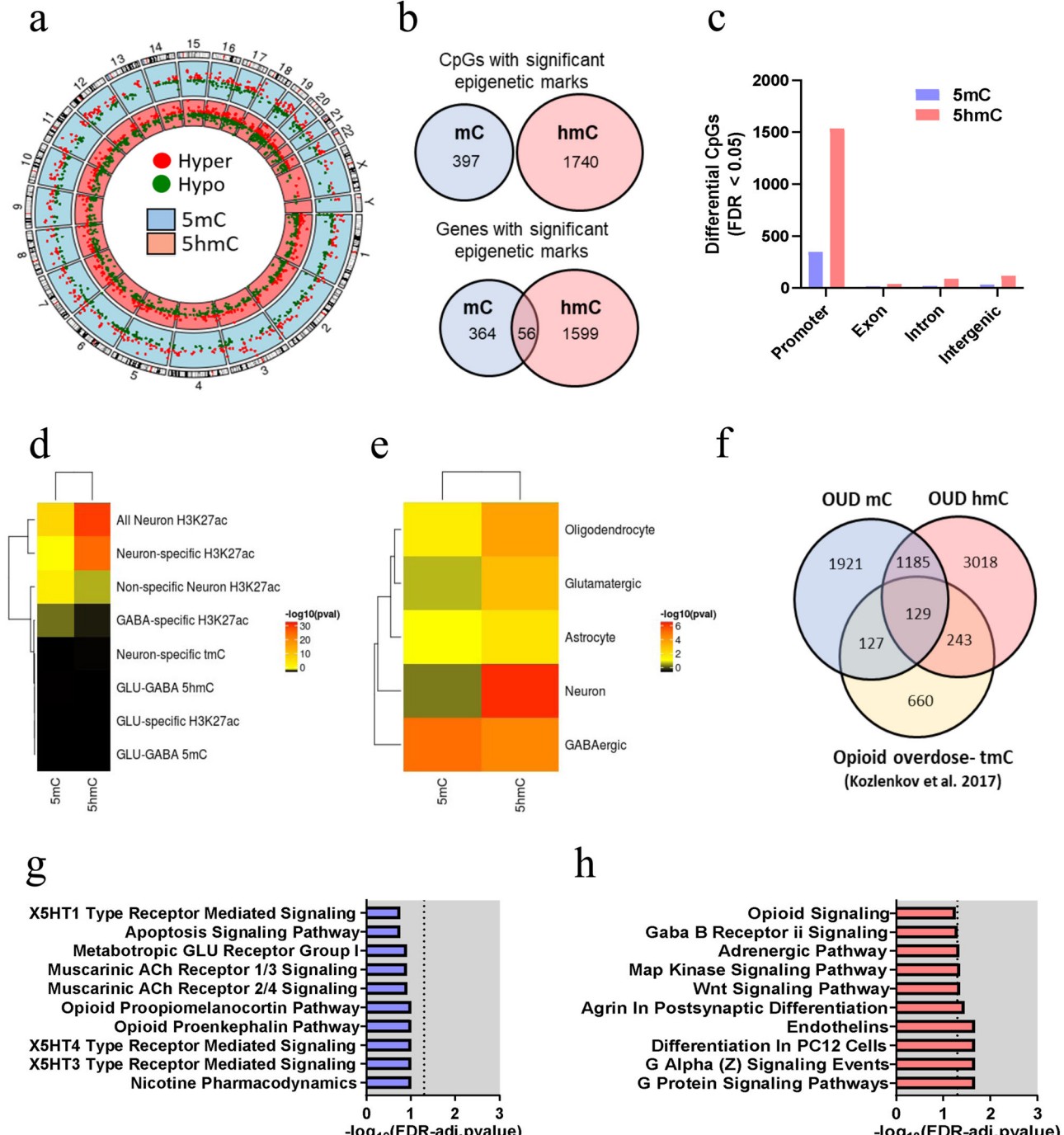

**Fig. 2 | Differential 5mC and 5hmC marks of OUD. a** Distribution of differential sites into Chromosomal location of hyper- and hypomethylation for 5mC and 5hmC. **b** Comparison between differential 5mC and 5hmC sites for CpG and gene, showing a gene overlap of 56 annotated genes. **c** Distribution of differential CpGs across genomic loci showed the promoter region with the highest number of 5mC and 5hmC differential CpGs. **d**, **e** Enrichment analysis for neuronal regions showed

that 5mC and 5hmC differential CpGs were mostly enriched for neuronal H3K27ac. **f** Comparison of our differential 5mC and 5hmC CpGs with differential 5mC markers detected in the OFC of opioid overdose cases (Kozlenkov et al.) showed an overlap of 129 genes. **g** Gene ontology analysis for OUD-associated 5mC genes. **h** Gene ontology analysis for OUD-associated 5hmC genes. Enrichment analyses were calculated using Fisher exact tests.

OUD-associated modules, we identified *DDX31* in the turquoise3 module. In the wheat1 module, we found one differential 5mC annotated gene that was antisense to the *JDP2* gene. No differential CpGs were enriched in the aquamarine1, ghostwhite, lightpink2, or turquoise1 modules. In the 5hmC OUD-associated modules, we identified three genes (*RPL21*, *RPS21*, and *ALYREF*) with differential 5hmC CpGs enriched in the navajowhite3 module. In the thistle1 module, we detected differential 5hmC CpGs in the *ARPC1B*, *DIMT1*, *PTBP1*, and

*CCT2* genes. The rosybrown1 module showed enrichment of differential CpGs in the genes *RPLP2*, *EIF3B*, *CCT2*, *NDUFB10*, and *ELAVL1*. Lastly, we identified two genes, *SRSF7* and *POLG2*, with differential CpGs enriched in the orange4 module.

**GWAS enrichment analysis**
We conducted a GWAS enrichment analysis to further assess the functionality of the identified differential 5mC and 5hmC genes using

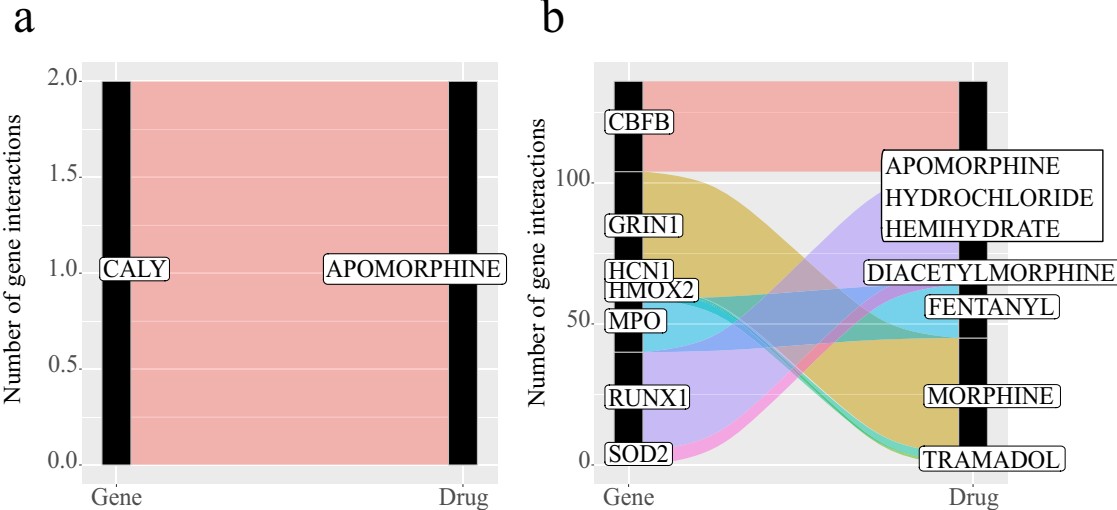

**Fig. 3 | Drug interaction analysis of the annotated genes with OUD-associated differential 5mC and 5hmC CpG sites.** The Alluvial plot shows the number of described opioid-related drug interactions (right axis) for each annotated gene (left axis) with **a** OUD-associated differential 5mC CpG sites, and **b** OUD-associated differential 5hmC CpG sites. Band color represents each of the annotated genes with interactions with opioid-related drugs. Bandwidth indicates the number of interactions with opioid-related drugs.

deposited datasets in the FUMA_GWAS. No GWAS enrichment was found for differential 5mC genes. However, differential 5hmC genes showed enrichment for skeletal domain, primarily related to height and movement structure (adj. *p* value range: 4.25E-12 to 1.86E-09; Supplementary Data 14, Supplementary Fig. 6A). We also observed GWAS enrichment in the psychiatric domain, including hyperkinetic disorders, pervasive developmental disorders, bipolar affective disorder, sleep functions, temperament and personality functions, and schizophrenia at the subdomain level (adj. *p* value range: 5.03e-7 to 0.044). Additionally, we tested the OUD GWAS enrichment of the differential 5mC and 5hmC genes using data from a recent OUD GWAS in individuals of African and European American ancestry[27]. For 5mC, we found five overlapping genes (Supplementary Data 8), but we did not observe a significant enrichment (odds ratio = 1.4371, *p* value = 0.2758). However, for the differential 5hmC genes, we observed a significant enrichment (odds ratio = 1.8757, *p* value = 0.0028) with 27 overlapped genes (Supplementary Data 8).

GWAS enrichment analysis was also performed for all 14 identified OUD-associated co-methylation modules (Supplementary Data 15, Supplementary Fig. 6B). We observed a similar enrichment pattern to the differential 5hmC CpG sites, with a higher enrichment of sites associated with the skeletal domain (adj. *p* value range: 2.96E-30 to 0.018). Additionally, for the 5hmC OUD-associated modules navajowhite3, orange4, thistle1, and rosybrown1, as well as the 5mC OUD-associated module lightpink2, we detected enrichment of genes related to the cognitive function domain (adj. *p* value range: 0.00090 to 0.046). Furthermore, both the 5mC and 5hmC OUD-associated modules showed enrichment in the psychiatric domain, including various subchapter levels such as sleep functions, depressive episode, schizophrenia, bipolar affective disorder, temperament and personality functions, recurrent depressive disorder, failure of genital response, hyperkinetic disorders, mental and behavioral disorders due to use of tobacco, and mental and behavioral disorders due to use of alcohol (adj. *p* value range: 2.83E-19 to 0.047).

**Bulk OFC gene expression analyses**

A differential gene expression (DEG) analysis was performed on 38 bulk tissue samples (OUD = 12, non-OUD = 26), using the same samples as in the 5mC/5hmC analyses. After Bonferroni correction, only one gene, Hemoglobin Subunit Beta (*HBB*), showed differential expression

in association with OUD (*p* value = 1.58E-06, Supplementary Data 16, Fig. 6a). We also conducted a correlation analysis between bulk gene expression and neuronal-specific 5mC/5hmC levels to explore the effect of these epigenetic marks on gene regulation. We identified 64 differential 5mC CpGs (Supplementary Data 17) and 257 differential 5hmC CpGs (Supplementary Data 18) that were correlated with gene expression levels. Notably, the two genes with the highest correlation between differential 5mC/5hmC and gene expression were the long non-coding RNA *LINC01002* and *TMIGD3*.

## Discussion

This study represents a comprehensive parallel investigation of neuronal 5mC and 5hmC in the context of OUD in the human brain. Our main findings suggest an important regulatory role of 5hmC in OUD, as evidenced by a higher number of differential CpGs and OUD-correlated modules identified, along with a stronger concordance with gene expression patterns. The dysregulated 5mC and 5hmC CpGs and modules showed enrichment for neuronal function and development and exhibited nominal associations with opioid signaling. Moreover, the OUD-associated 5hmC marks showed enrichment for the G protein signaling pathway, Wnt signaling, and psychiatric domains. The drug interaction analysis revealed opioid interactions with one gene with differential 5mC and seven genes with differential 5hmC, including *RXN1*, *GRIN1*, and *CBFB*. Additionally, we identified *HBB* as a differentially expressed gene in OUD and observed correlations between 5mC/5hmC and gene expression patterns, indicating the role of these epigenetic marks on gene regulation.

Genome-wide differential CpG analysis identified 397 CpGs for 5mC and 1740 CpGs for 5hmC associated with OUD. Our 5mC and 5hmC findings showed overlap with other recently published 5mC studies of opioid-related traits. In the human postmortem brain, we observed a replication in a previous study assessing 5mC in the postmortem OFC of individuals that experienced heroin abuse and died from overdose[12], in which 327 genes overlapped (127 with 5mC; Supplementary Data 6). When considering recent 5mC studies in the dorsolateral prefrontal cortex of opioid-related traits[13,14], we identified two overlapped genes (*SMARCA4*; *JUN*) identified as potential transcriptional regulators in a multi-omics study of OUD in postmortem brain tissue[14]. Expanding to a multi-tissue comparison, we observed overlap between our 5mC and 5hmC annotated genes in studies

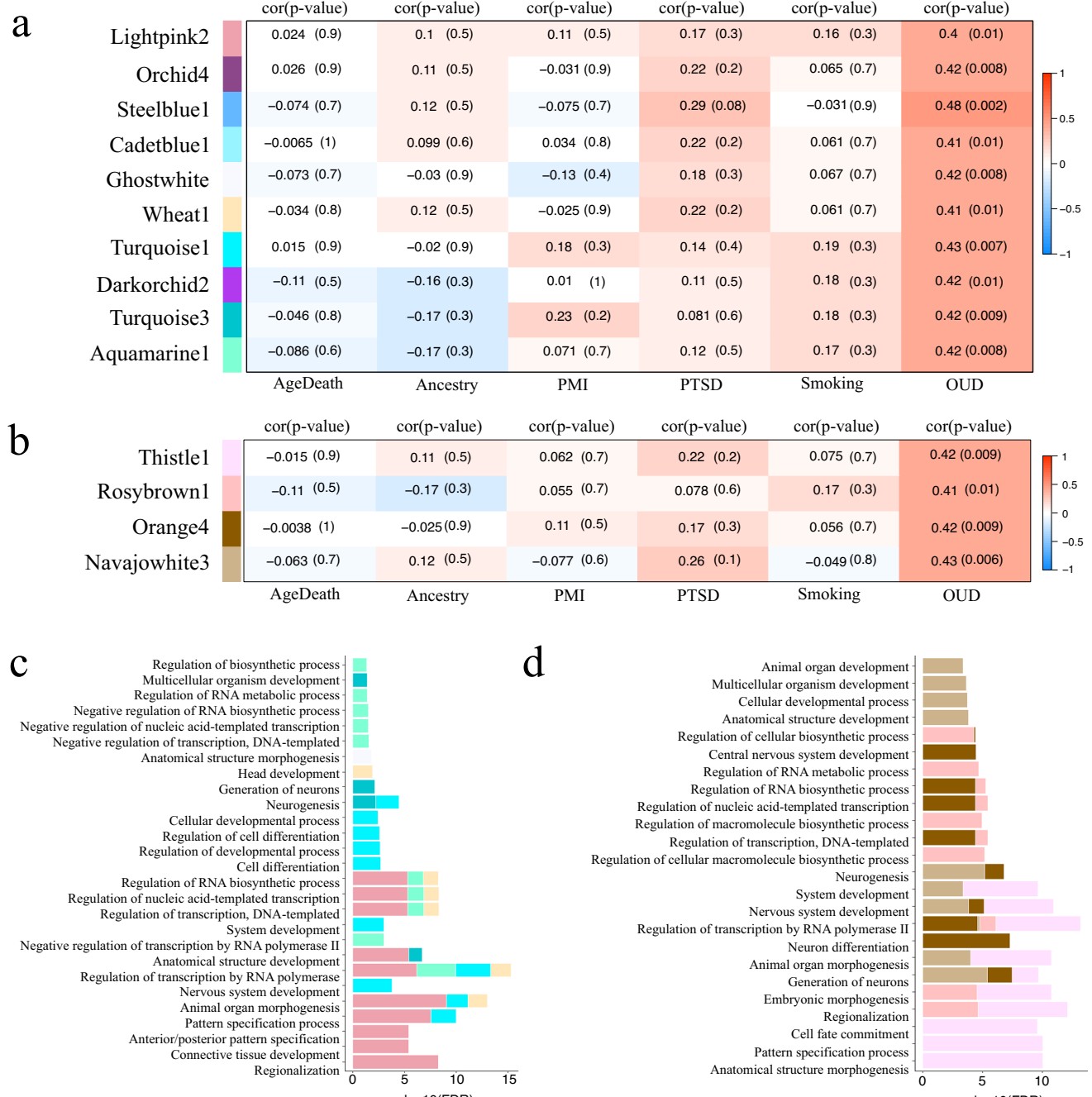

**Fig. 4 | Co-methylation of 5mC and 5hmC sites. a** 5mC-significant modules associated with OUD (correlation ≥ |0.4|, *p* value < 0.05). **b** 5hmC-significant modules associated with OUD (correlation > ≥|0.4|, *p* value < 0.05). Correlations were calculated using weighted Pearson correlation, and *p* value using student asymptotic statistics. **c** GO enrichment analysis for 5mC-significant modules (top 10 terms). **d** GO enrichment for 5hmC-significant modules (top 10 terms).

evaluating placenta[23,24], saliva[22], and blood[25]. Most of the convergence was observed with 5mC studies in the placenta evaluating neonatal opioid withdrawal syndrome[23,24]. When examining *OPRM1*, one of the most reported associated genes in the opioid literature, we found that this gene was identified in our 5mC co-methylation analysis, specifically in the OUD-associated lightpink2 module. Taken together, our findings are consistent with the literature identifying previously reported associations, but also potentially unknown ones, that may play an important regulatory role in OUD in the human brain.

When evaluating the enrichment of the differential 5mC and 5hmC genes, only 5hmC showed significant enrichment after multiple testing corrections in several pathways, some of which have been previously implicated with OUD, including Wnt signaling[28] and

G-protein signaling. Morphine and other opioids are known to activate μ-opioid G protein-coupled receptors to elicit tolerance and dependence[29]. We also observed nominal enrichment in both 5mC and 5hmC for opioid-related pathways: 5mC CpGs were enriched in the opioid proenkephalin pathway and opioid proopiomelanocortin pathway and 5hmC CpGs in opioid signaling. Moreover, the GWAS enrichment analysis of 5hmC differential genes identified enrichment for an OUD GWAS[27].

For the co-methylation analysis, we observed the enrichment of several pathways previously associated with opioids, including Reactome pathway Pre-NOTCH Transcription and Translation, Wnt signaling pathway, inhibition of voltage-gated Ca2+ channels via inhibition of G protein-gated potassium channels, class B/2 (secretin family

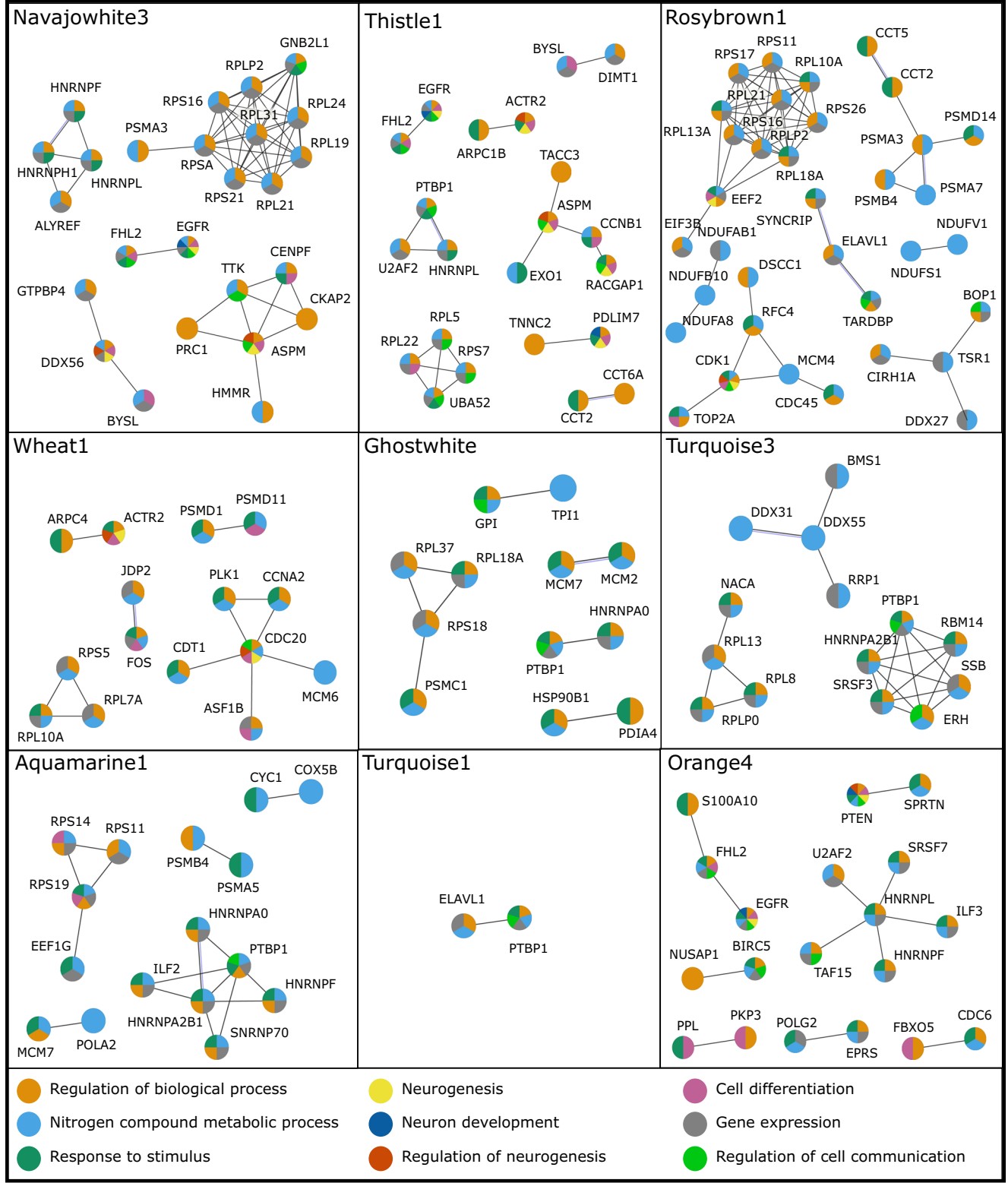

**Fig. 5 | Network of 5mC- and 5hmC-significant modules associated with OUD.** The figure shows the PPI analyses for 9 modules detected with a significant association with OUD in 5mC (wheat1, ghostwhite, turquoise3, aquamarine1, turquoise1) and 5hmC (navajowhite3, orange4, rosybrown1, thistle1). The colors represent a different biological process in which the modules were enriched. In 5hmC OUD-associated modules, we observed a higher number of annotated genes involved in neurogenesis and regulation of neurogenesis.

receptor), and insulin receptor signaling cascade. NOTCH signaling has been implicated in the adaptation to chronic morphine exposure and proposed as a potential target for pain management[30]. Voltage-gated Ca2+ channels have been involved in the release of pain

neurotransmitters[31]. Wnt signaling pathway was enriched in the differential 5mC analysis and has been linked to opioid-related withdrawal symptoms in mice[28]. Class B/2 (secretin family receptor) was also identified in a co-expression analysis of maternal exposure to

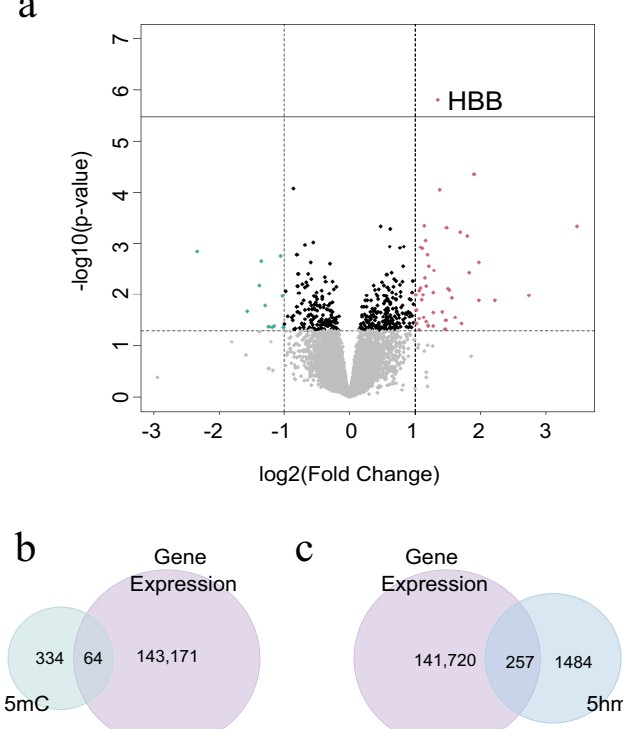

**Fig. 6 | Differential expression analysis of OUD. a** Volcano plot (threshold: most significant, $10^{-5}$). The graph identifies *HBB* as a differentially expressed gene associated with OUD. **b** Venn diagram for the methylated sites correlated with gene expression against differential 5mC CpG sites shows an overlap of 64 genes. **c** Venn diagram for the hydroxymethylated sites correlated with gene expression against the differential 5hmC CpG sites shows an overlap of 257 genes.

oxycodone[32]. Chronic exposure to opioid drugs, such as morphine, is known to inhibit the insulin signaling pathway, leading to alterations in glucose homeostasis, and an increased risk of developing insulin resistance and associated comorbidities[33,34]. The PPI analysis for the modules showed evidence of co-expression in genes involved in neurogenesis pathways, which corroborates previous studies showing that opioids decrease neurogenesis by inhibiting cell division, mainly through blocking the S phase[35]. It also revealed an interaction with one differential 5hmC annotated gene, *CCT2*, present in the navajowhite3 and rosybrown1 modules, while no annotated genes were observed for 5mC. *CCT2* is a chaperonin with a role in hypoxia in colorectal cancer[36]. Our findings unveil epigenetically dysregulated gene networks in OUD involving the Wnt, immune, and pain signaling pathways, as well as neurogenesis.

Our gene-drug interaction analyses with opioids revealed a higher number of associations with differential 5hmC marks, suggesting a more significant functional role of 5hmC in OUD. Evaluating the individuals who died from drug intoxication, we observed in the gene-drug interaction analysis that only one of the opioids (fentanyl) was detected at the time of death. However, further investigation is needed to clarify whether the effects of gene methylation/hydroxymethylation are associated with drug intoxication. *CALY* was annotated in the 5mC differential genes, and it has been previously associated with smoking initiation in Chinese and American populations[37]. In the 5hmC differential genes, we identified the gene coding GRIN1, known to regulate μ-Opioid activity;[38] *SOD2*, previously described as associated with the risk of heroin dependence in an Iranian population ($n = 1241$);[39] and *HCN1*, in which the protein is shown to be required for the activation of the μ opioid receptor[40]. We also identified opioid interactions with the gene that codes for RUNX1, known to bind in the promoter region of

the *ADRA1A* gene. Hypermethylation in the *ADRA1A* gene has been previously associated with OUD in a candidate gene study in the Han Chinese population[41]. *HMOX2*, *MPO*, and *CBFB*, also identified, have not been previously described in OUD. Lofexidine and tizanidine, drugs previously used for the treatment of OUD symptoms, were observed in 5hmC annotated genes with described interactions with *ADRA2A*, a gene involved in the release of neurotransmitters. Our results demonstrate that epigenetic marks, mostly 5hmC, functionally interact with opioids and pinpoint promising targets for OUD treatment.

To better understand the impact of 5mC and 5hmC alterations on gene expression, we conducted a DEG analysis followed by integration with 5mC and 5hmC data. Differential analysis of RNA sequencing data from bulk tissue identified the *HBB* gene as associated with OUD. HBB has been reported in neurodegenerative neurons associated with Parkinson's[42], and Alzheimer's[43]. Neurodegeneration has been observed in opioid use studies[44,45]. Further, hemoglobin acts as an oxygen-storage molecule in hypoxia, which is suggested to be induced by opioids in rodents[46], indicating a potential increase in brain hypoxia in individuals with OUD. Our findings suggest that *HBB*, an important gene involved in the regulation of oxygen homeostasis, is also associated with OUD. However, future studies are needed to disentangle this association. The correlation analysis between 5mC/5hmC and gene expression levels detected an overlap of 64 differential CpG sites for 5mC and 257 for 5hmC. Additionally, a higher correlation was observed for 5hmC CpGs in the *TMIGD3* (Transmembrane and Immunoglobulin Domain Containing 3) gene. TMIGD3 can act as a repressor of NF-κB[47], an essential protein that regulates inflammation and is altered in individuals with OUD[48,49]. These results suggest a higher impact of 5hmC than 5mC on gene regulation in OUD.

This study has several strengths. In this study, we assess 5hmC in parallel with 5mC in OUD. Most studies have used bulk tissue, where molecular changes may be obscured, as also evident in our bulk gene expression results. We performed FANS-sorted neuronal nuclei, allowing us to conduct cell type-specific mapping of 5mC/5hmC in the human OFC. Leveraging RRoxBS resulted in three times more CpG coverage than the latest and most used microarray technology (Infinium MethylationEPIC Array). We also conducted RNA sequencing analysis for the same subjects and integrated it with 5mC and 5hmC data to evaluate the effects of these epigenetic modifications on gene transcription. 5mC and 5hmC patterns were assessed using two analytical approaches: CpG site and co-methylation analysis, enabling the identification of associations not only at the single-CpG loci level but also within epigenetically regulated gene networks. We also integrated our epigenetic findings with GWAS data, showing an association of OUD-associated 5hmC genes and modules with psychiatric disorders, including traits related to substance use disorders.

The study is limited by the small sample size, but it is comparable to similar work in the human postmortem brain. Notably, our cohort is heterogeneous in terms of comorbidities and drug intoxication at the time of death. For example, all individuals with OUD were comorbid with PTSD. We addressed this issue by adding PTSD as a covariate in the model (PTSD is present in 50% of the non-OUD group) and replicated previous work assessing 5mC in the OFC of individuals that experienced heroin abuse[12]. Furthermore, some of the samples from the non-OUD group were positive for opioid intoxication at the time of death. RNA sequencing data were conducted in bulk tissue, which may impede our ability to identify OUD DEGs and directly evaluate the impact of differential 5mC/5hmC on gene expression in a cell type-specific manner. Another limitation is the sole inclusion of males, mostly of European ancestry. Future work will expand to female subjects and other ancestries to identify potential sex- and population-specific effects. Further, additional comorbidities (e.g., MDD and other SUDs) should be also examined. This study evaluated 5mC and 5hmC at CpG sites; future studies should evaluate the role of 5mC and 5hmC in non-CpG sites in the context of OUD, which have been suggested to

play a crucial role in the development of neuropsychiatric diseases[50]. Given the enrichment of differential 5mC and 5hmC genes and modules in GWAS studies, future work should evaluate potential methylation/hydroxymethylation quantitative trait loci to assess whether differential 5mC/5hmC is influenced by genetic background. Although the enrichment of GWAS signals may be indicative of potential causal effects of the OUD-associated 5mC/5hmC marks, research in model organisms (e.g., animal models, human induced pluripotent stem cells, brain organoids) could help confirm whether the identified marks are the cause or consequence of OUD.

In summary, our findings identified 5mC and 5hmC dysregulation in OFC neurons from individuals with OUD. The results suggest that 5hmC plays an important role in OUD, as shown by the magnitude and functionality of differential marks and networks, as well as its potential impact on gene expression patterns. We have identified clinically relevant pathways, such as NOTCH signaling processes, Wnt signaling, and G protein signaling pathways, as well as enrichment for psychiatric domains, including substance use disorders. Lastly, multiple gene-drug interactions between differential 5mC/5hmC and opioids were identified, revealing targets of well-known and potential OUD treatments. Our study supports the important role of 5hmC on OUD and demonstrates a multi-omics dysregulation of OUD in the human OFC, identifying well-known and potential targets that may inform prognostic and treatment efforts in the future.

## Methods

### Sample description
Postmortem human brain specimens were obtained from the National Post-Traumatic Stress Disorder (PTSD) Brain Bank[51] (NPBB), a brain tissue repository at the U.S. Department of Veterans Affairs (VA). Consent was obtained next-of-kin. In addition, brain tissue samples were collected from the orbitofrontal cortex (OFC; Brodmann Area 11). The cause of death in the non-OUD group included natural causes ($n = 13$, 34.21%), suicide ($n = 5$, 13.16%), accident causes ($n = 7$, 18.42%) mainly by body injuries and gunshots, and alcohol and drug intoxication ($n = 3$, 7.89%, including two opioid-related intoxications). None of the individuals in the non-OUD group had a history of OUD diagnosis. All individuals in the OUD group were diagnosed with OUD and died from drug and/or alcohol intoxication, including, but not limited to, opioids. Table 1 presents the demographics and clinical characteristics of the study cohort. Diagnoses were conducted using two approaches: an antemortem assessment protocol for antemortem donors and a postmortem diagnostic assessment for postmortem donors, as described by Friedman and colleagues[51]. This study was approved by the Institutional Review Board Committees of the Department of Veterans Affairs and Yale School of Medicine.

### Fluorescence-activated nuclei sorting
For each specimen, OFC tissue (100–200 mg) was lysed in homogenization buffer (0.1% Triton, 0.32 M sucrose, 5 mM CaCl$_2$, 3 mM MgCl$_2$, 10 mM Tris-HCl) on ice in a glass Dounce homogenizer. The homogenized samples were then filtered through a 40 μM cell strainer, loaded onto a 1.8 M sucrose cushion, and ultracentrifuged (SW-41 rotor, Beckman Coulter, Brea, California, USA) at 24,000 rpm for 1 hour at 4 °C. Nuclei were resuspended in 0.5% bovine serum albumin and stained for 30 min at 4 °C with Anti-NeuN-PE (Millipore-Sigma, FCMAB317PE). Before sorting, 4',6-diamidino-2-phenylindole (DAPI) was added as a nuclei label, and samples were again filtered through a 40 μm strainer. Fluorescence-activated nuclear sorting (FANS) procedures were carried out at the Icahn School of Medicine Flow Cytometry CoRE using a BD 5-laser cell sorting system.

### DNA extraction
Between 0.5–1 M NeuN+ nuclei were collected via FANS and processed for DNA extraction. First, sorted nuclei were pelleted by centrifugation

at 1500 × g for 15 min at 4 °C. Next, the supernatant was aspirated down to 500 μL and 50 μL proteinase K (Cat. #69504, Qiagen, Valencia, CA) and 20 mg/mL RNAse A (Cat. #12091021; Thermo-Fischer, Waltham, MA) were added. The samples were processed following the manufacturer's protocol of the DNeasy Blood and Tissue Kit (Cat. #69504, Qiagen). Finally, the eluted samples were concentrated to a final volume of 20 μL using the Zymo Genomic DNA Clean and Concentrator-10 kit (Cat. #D4010, Zymo Inc., Irving CA) and stored at −80 °C.

### Reduced representation oxidative bisulfite sequencing (RRoxBS)
RRoxBS was carried out at the Weill Cornell Epigenomics Core (New York, NY). Briefly, two libraries were prepared from 400 ng of DNA to analyze methylation and hydroxymethylation at CpG dinucleotides in each subject using the NuGEN Ovation RRoxBS Methyl-Seq library preparation kit. One library underwent bisulfite treatment to convert unmethylated cytosines to uracils (BS), while the second library underwent oxidation prior to bisulfite sequencing (oxBS) to convert both unmethylated and hydroxymethylated cytosines to uracils. The prepared libraries were multiplexed and pooled for single-end 1 × 50 bp sequencing to a mean depth of 42.7 ± 1.5 (μ ± SEM) million reads per library on an S4 flow cell using the Illumina NovaSeq6000 system.

### Bioinformatic analysis
Raw data were processed by the Weill Cornell Epigenomics Core (New York, NY). The Bismark bisulfite read mapper[52] was used to map sequencing reads to the CRCh38 human genome and detect bisulfite treatment of converted and unconverted cytosines. Two libraries were sequenced for each sample: 1. Bisulfite sequencing (BS), which converts unmethylated and unhydroxymethylated cytosines into thymine; and 2. Oxidative BS (oxBS), which allows the conversion of hydroxymethylated cytosines into thymine. Beta values (% unconverted cytosines) are calculated for 5hmC as the difference in methylation at each CpG site between BS and oxBS libraries (e.g., if $\beta = 100\%$ in the BS library and $\beta_{5mC} = 60\%$ in the oxBS library, then 5hmC = 40%). Analyzed CpGs were filtered for a minimum of 10× coverage in all subjects and were normalized for coverage variability.

The annotation of CpGs overlapping with neuronal active enhancer regions (H3K27ac) was performed using previously published ChIP-seq datasets[53] generated from glutamatergic and GABAergic cell types of the human orbitofrontal cortex[53]. In addition, non-neuronal enhancers were taken from Nott et al.[54]. Cell type-specific gene markers used for enrichment analysis were derived from McKenzie et al.[55]. Regions with total differential methylation (tmC=5mC+5hmC) between NeuN+ and NeuN− populations (>50% difference in tmC) of the human prefrontal cortex from Rizzardi et al.[20] were utilized for comparison with our NeuN+ dataset.

### Differential analysis of methylation and hydroxymethylation
Differential analysis of 5mC and 5hmC was performed at single-CpG resolution with the methylkit R package[56]. Logistic regression with correction for overdispersion and chi-squared significance testing was applied, and the analysis was benchmarked to achieve the best balance of sensitivity and specificity[57]. P values were calculated using the logistic regression and adjusted to q value using Sliding Linear Model (SLIM)[58], which incorporates simulated data to account for data structure and hypothesis. Q value thresholds were applied at 0.05. Bonferroni-adjusted findings are also reported in Supplementary Data 1 and 2, using a more stringent threshold (5mC: $p = 2.71\text{E-}8$; 5hmC: $p = 3.02\text{E-}8$) to assess the results. Additionally, we evaluated these epigenetic marks summed over neuronal H3K27ac regions with ≥3 CpGs at ≥10× coverage. Age, ancestry, postmortem interval (PMI), cigarette smoking, and PTSD were included as a priori covariates. The

5mC and 5hmC analyses were performed separately, and significance was set at false discovery rate (FDR)-adjusted *p* value < 0.05.

To investigate functional gene sets enriched for differential CpGs, gene ontology (GO) analysis was conducted using the methylGSA Bioconductor package[59]. Gene lists from the NCATS BioPlanet, Gene Ontology Consortium, and Kyoto Encyclopedia of Genes and Genomes (KEGG) databases were analyzed. Furthermore, enrichment of differentially 5mC and 5hmC markers for genomic regions and features were assessed using one-sided Fisher's exact tests.

### Differential 5mC and 5hmC annotation
Annotation of the closest Ensembl transcript ID of the differential sites was performed using the Genomation[60] R package. The annotation considered any distance between the site and the transcription start site (TSS). Gene symbols were annotated using the biomaRt[61] R package. In addition, a complimentary annotation was conducted using the UCSC genome browser[62] to identify the gene region location. A threshold of 1500 bp[63] was applied ("Position to the Nearest gene ID" in Supplementary Data 1 and 2).

### Co-methylation analysis
Co-methylation analysis was conducted separately for 5mC and 5hmC using the WGCNA R package[64]. Quality control was performed to remove sites with missing values. In the co-methylation analysis, 1,835,925 sites were retained for 5mC and 1,638,378 sites for 5hmC. Age, ancestry, PMI, PTSD, and smoking status were included as covariates using the empiricalBayesLM function from the WGCNA package. The blockwiseModules function was used for module detection, with a soft-threshold power of 5 and TOMType set as signed for both analyses. The eigengene calculated for each module was used for correlation analysis with OUD. Correlation with an absolute value greater than |±0.4| and *p* value < 0.05 were considered significant. Enrichment analysis was performed using AmiGO[65], considering Gene Ontology (GO) for biological processes and Reactome pathways.

The nearest gene IDs were subjected to protein-protein interaction (PPI) using STRING[66]. Genes from the top 20 biological processes GO terms, after PPI analysis, were selected to reduce the number of nodes. For the thistle1 module, genes from the top 10 biological processes GO terms were used. The parameters were set to show evidence based on co-expression data, with a minimum required interaction score of 0.9.

### RNA sequencing
RNA was extracted from bulk tissue medial OFC (BA 11) for all samples[67]. Briefly, 20 mg of tissue was isolated using the RNeasy Mini Kit (Qiagen) with genomic DNA elimination. RNA integrity and concentration were assessed using a Bioanalyzer (Agilent) and rRNA was depleted using Ribo-Zero Gold Kit (Illumina). Libraries were constructed using the SMARTer® Stranded RNA-seq Kit (Takara Bio) and sequenced at 75 bp paired-end on an Illumina HiSeq4000. RIN values for the non-OUD and OUD groups were similar (mean non-OUD = 7.64 ± 1.14 and mean OUD = 8.38 ± 0.50).

### Differential gene expression
FASTQ files were mapped using STAR (v.2.5.3a) and counted using featureCounts (v.1.5.3). Further, it was annotated using the GTF file downloaded from ENSEMBL (release 79, GRCh38) and the biomaRt package in R. Differentially expressed genes (DEGs) were calculated using the DESeq2 package in R[68]. The statistical framework enabled the calculation of $\log_2$ fold-change values ($\log_2$FC) for each gene in the PTSD + OUD and PTSD + CON raw count data. This model considered the following covariates to measure the association with opioid use: age, RNA integrity number (RIN), PTSD diagnosis, and smoking status. Genes with zero expression were dropped. Differentially expressed genes (DEGs) were defined using an FDR-adjusted *p* value < 0.05 (Benjamini–Hochberg).

### Correlation analysis between gene expression and 5mC/5hmC
The correlation between gene expression and 5mC/5hmC data was calculated using the MatrixEQTL package[69]. Pearson correlation analysis was performed in cis and trans. Cis was defined using the default parameters from the Matrix_eQTL_main function, that is, within the threshold of 1e-6 bp from the starting or ending of the annotated gene, while *trans* included everything outside that threshold. The raw count data from the RNA-seq analysis was normalized using variance stabilization transformation. Covariates included age, ancestry, RIN, PMI, PTSD, smoking status, and three surrogate variables. Significance was defined as an FDR-adjusted *p* value < 0.05.

### GWAS enrichment analysis
GWAS enrichment analysis was conducted to test the probability of overlapping genes between the 5mC/5hmC analysis and published GWAS findings. The analysis was performed using the FUMA online website[70], which applies a hypergeometric test for enrichment. Ensembl IDs were transformed to Entrez IDs using David[71] and submitted to FUMA's GENE2Function web tool. Enrichment results were merged with domains and traits reported by the GWAS Atlas using in-house scripts. Additionally, GWAS enrichment specific to OUD was conducted using data from a recently published GWAS in European and African Populations[27] For this, we used Fisher's exact test and a similar background to the Metascape platform (30,182 genes)[70]. Significance was defined as FDR < 0.05 in both approaches.

### Drug interaction analysis
Drug interaction analysis was conducted for the annotated genes of the differential 5mC and 5hmC CpGs using the Drug Gene Interaction Database (DGIdb; https://www.dgidb.org)[72]. From the gene–drug interactions identified, we further evaluated those in 15 opiates: apomorphine, codeine, diacetylmorphine, hydrocodone, methadone, morphine, oxycodone, oxymorphone, tramadol, methadone, propoxyphene, fentanyl, hydromorphone, heroin, and levorphanol. In addition to opioids, we also evaluated other drugs that showed interactions with the same annotated genes.

### Reporting summary
Further information on research design is available in the Nature Portfolio Reporting Summary linked to this article.

## Data availability

We have published our repository at https://doi.org/10.5281/zenodo.7942472 and our data[73] at https://doi.org/10.5281/zenodo.7958290 (GEO: GSE235818). Source data are provided with this paper.

## Code availability

The scripts used in the manuscript may be accessed on the montalvoortizlab GitHub[74] (https://github.com/montalvoortizlab/RRoxBS).

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

## Acknowledgements

This work is supported by the U.S. Department of Veterans Affairs via 1IK2CX002095-01A1 (J.L.M.O.) and NIDA R21DA050160 (J.L.M.O.). We thank Dr. Sarah Beck for editing the manuscript.

## Author contributions

G.R. and S.T.N. conducted the analyses and drafted the manuscript. J.J.M.M. performed the GWAS enrichment and correlation analyses between gene expression and 5mC/5hmC data and assisted in the manuscript writing (Methods section). D.L.N.R. performed the annotation analyses of the differential 5mC/5hmC data and assisted in manuscript writing (Methods section) and revisions. J.W. and M.J.G. performed the RNA-seq analysis and assisted in manuscript writing (Methods section). J.H.K., J.G., and T.S.B.R.G. provided feedback and revised the manuscript. Y.H. conceived and designed the study and revised the manuscript. J.L.M.O. conceived, designed, and coordinated the study, and supervised the manuscript preparation and revision. All authors contributed to and approved the final manuscript.

## Competing interests

The following competing interests for J.H.K.: (1) Consultant: Note: The Individual Consultant Agreements listed below are <$10,000 per year: AstraZeneca Pharmaceuticals; Biogen, Idec, MA; Biomedisyn Corporation; Bionomics, Limited (Australia); Concert Pharmaceuticals, Inc.; Heptares Therapeutics, Limited (UK); Janssen Research & Development; L.E.K. Consulting; Otsuka America Pharmaceutical, Inc.; Spring Care, Inc.; Sunovion Pharmaceuticals, Inc.; Takeda Industries; Taisho Pharmaceutical Co., Ltd; Scientific Advisory Board; Bioasis Technologies, Inc.; Biohaven Pharmaceuticals; Blackthorn Therapeutics, Inc.; Broad Institute of MIT and Harvard; Cadent Therapeutics; Lohocla Research Corporation; Pfizer Pharmaceuticals; Stanley Center for Psychiatric Research at the Broad Institute; (2) Stock: ArRETT Neuroscience, Inc.; Blackthorn Therapeutics, Inc.; Biohaven Pharmaceuticals Medical Sciences; Spring Care, Inc. Stock Options: Biohaven Pharmaceuticals Medical Sciences; (3) Income Greater than $10,000: Editorial Board Editor—Biological Psychiatry; Patents and Inventions: Seibyl JP, Krystal JH, Charney DS. Dopamine and noradrenergic reuptake inhibitors in the treatment of schizophrenia. US Patent #:5,447,948. 5 September 1995; Vladimir, Coric, Krystal, John H, Sanacora, Gerard—Glutamate Modulating Agents in the Treatment of Mental Disorders US Patent no. 8,778,979 B2 Patent Issue Date: 15 July 2014. US Patent Application No. 15/695,164: Filing Date: 09/05/2017; Charney D, Krystal JH, Manji H, Matthew S, Zarate C. —Intranasal Administration of Ketamine to Treat Depression United

States Application No. 14/197,767 filed on 5 March 2014. United States application or Patent Cooperation Treaty (PCT) International application no. 14/306,382 filed on 17 June 2014; Zarate, C, Charney, DS, Manji, HK, Mathew, Sanjay J, Krystal, JH, Department of Veterans Affairs "Methods for Treating Suicidal Ideation", Patent Application No. 14/197.767 filed on 5 March 2014 by Yale University Office of Cooperative Research; Arias A, Petrakis I, Krystal JH. —Composition and methods to treat addiction; Provisional Use Patent Application no.61/973/961. April 2, 2014. Filed by Yale University Office of Cooperative Research; Chekroud, A., Gueorguieva, R., & Krystal, JH. "Treatment Selection for Major Depressive Disorder" [filing date 3rd June 2016, USPTO docket number Y0087.70116US00]. Provisional patent submission by Yale University; Yoon G, Petrakis I, Krystal JH. —Compounds, Compositions, and Methods for Treating or Preventing Depression and Other Diseases. U.S. Provisional Patent Application No. 62/444,552, filed on 10 January 2017 by Yale University Office of Cooperative Research OCR 7088 US01; Abdallah, C, Krystal, JH, Duman, R, Sanacora, G. Combination Therapy for Treating or Preventing Depression or Other Mood Diseases. U.S. Provisional Patent Application no. 047162-7177P1 (00754) filed on 20 August 2018 by Yale University Office of Cooperative Research OCR 7451 US01. Dr. Gelernter is named as an inventor on PCT patent application #15/878,640 entitled: "Genotype-guided dosing of opioid agonists," filed on 24 January 2018 and issued on 26 January 2021 as U.S. Patent No. 10,900,082. J.G. is paid for editorial work for the journal "Complex Psychiatry." Other co-authors declare no competing interests.

## Additional information

## Traumatic Stress Brain Research Group

Victor E. Alvarez[7], David Benedek[8], Alicia Che[9], Dianne A. Cruz[10], David A. Davis[11], Matthew J. Girgenti[9,12], Ellen Hoffman[9], Paul E. Holtzheimer[12,13], Bertrand R. Huber[8], Alfred Kaye[9], John H. Krystal[9,12], Adam T. Labadorf[8], Terence M. Keane[12,8], Mark W. Logue[12,8], Ann McKee[8], Brian Marx[12,8], Mark W. Miller[12,8], Crystal Noller[12,13], Janitza Montalvo-Ortiz[9], William K. Scott[11], Paula Schnurr[12,13], Thor Stein[8], Robert Ursano[13], Douglas E. Williamson[10], Erika J. Wolf[12,8] & Keith A. Young[14]

[7]Boston University, Boston, MA, USA. [8]Uniformed Services University of the Health Sciences, Bethesda, MD, USA. [9]Yale University, New Haven, CT, USA. [10]Duke University School of Medicine, Durham, NC, USA. [11]University of Miami, Miami, FL, USA. [12]National Center for PTSD, Boston, MA, USA. [13]Geisel School of Medicine at Dartmouth, Hanover, NH, USA. [14]Texas A&M University, College Station, TX, USA.

