## [Peer Review File · Nature Communications]

Profiling Neuronal Methylome and Hydroxymethylome of Opioid Use Disorder in the Human Orbitofrontal CortexREVIEWER COMMENTS

Reviewer #1 (Remarks to the Author):

This is a study of methylation and hydroxymethylation in post-mortem brain tissue of subjects with an opioid use disorder (N=12) vs controls (N=26). This study is unique in that prior studies have largely assessed methylation, not hydroxymethylation. Moreover, the authors performed cell sorting to neurons from tissue derived from orbitofrontal cortex (although microdissection to orbitofrontal cortex not described), where most of the literature relies on bulk tissue.

The authors performed multiple types of analyses including differential methylation and hydroxymethylation in cases vs controls – single site, and then multiple types of enrichment analyses (pathway, overlap with H3K27ac, drug interaction pathways, co-methylation with OUD, enrichment in GWAS associated loci, and overlap with gene expression signal from bulk expression data the authors generated in this sample.

There are several noteworthy and interesting results, most importantly the more substantial signals for hydroxymethylation vs methylation. And the uniqueness of this study, a hydroxymethylation study in OUD, is another great strength.

Overall, the methodology is appropriate, with the exception of the group comparisons of covariates (table 1), which should be done by Fisher's Exact test. The sample is small, but the study is novel. The small sample size does make it difficult to assess the real impact of a study like this, but it certainly is novel.

There are some minor and major weaknesses that diminish enthusiasm for the manuscript. All are addressable and detailed below.

Major: There are substantial differences in the rates of several covariates, most notable in smoking (10/12 vs 13/26). Moreover, all OUD cases are also PTSD cases. While it is worth noting that the most important differences are included as covariates, which does reduce some concern, there are some issues with the presentation. First, given the small sample size, Fisher's exact test should be used to compare cases vs controls on demographic, psych characteristics. A cursory calculation shows that these groups differ significantly on PTSD and at a trend level on smoking. Second, the section in the text describing these covariates reports only the counts for the case group but not the control group, which is somewhat confusing.

Major: There is no discussion on drug use history or on the availability of toxicology screens for the subjects or the causes of death. There are obvious questions one would ask about the cases and controls. Were the subjects intoxicated at the time of death? Did the cases have a history of OUD or were they actively using opioids? Do the cases and controls differ on the cause of death (e.g., overdose, accident)? This information would help dramatically with interpretation of most of the results but most notable the drug interaction results.

Minor: According to the manuscript, the enrichment analyses relied on the “nearest gene”. Was there a maximum distance where a CpG was deemed “not in a gene”?

Minor: I think it is important to examine the overlap with existing literature, but the authors looked at one study. They should provide a rationale justifying that one study. It is also important to address whether or not the direction (hypo vs hyper) is the same as in those studies versus just reporting overlap in differential methylation/hydroxymethylation?

Minor: The entire manuscript could use a proofread. There are issues with noun verb agreement, especially in the discussion, open parentheses.

Minor: The labels on figures are not always informative.

Minor: The background literature is much more substantial than the one or two papers cited. I think it is important to include more of this to highlight that a substantial amount of work has been done on methylation but much less on hydroxymethylation

Minor: If there are not limitations on the words, the introduction should set the stage for this work more thoroughly. The authors should consider their target readership in editing this manuscript. It could be substantially more accessible.

Minor: The authors do not plan to share data other than summary statistics. This is problematic in the current data sharing environment.

Reviewer #2 (Remarks to the Author):

The overall goal of this study was to examine DNA methylation (5mC) and hydroxymethylation (5hmC) disturbances in neurons sorted from human postmortem brain samples of individuals with opioid use disorder (OUD) versus unaffected controls. The authors report identifying significant findings in both 5mC and 5hmC, correlation of sites with gene expression sequencing data from the same individuals and OUD-associated differentially expressed genes, and other results from additionally bioinformatic analyses. The manuscript is organized and well-written, and the findings are of importance to the field – though the importance of these findings is not always made apparent. Areas of concern are outlined below.

There are a handful of other studies examining DNA methylation and OUD or opioid use, however these studies and their findings are not currently discussed in the manuscript. Is there any overlap between the findings in this study and the previous studies? Perhaps the authors could perform a look-up replication of previous findings in their results?

The discussion currently reads as a re-statement of all the results. While, some of this is needed given the number of analyses, it is hard to know which findings are really important, why they are important them and how to interpret them within the context of what we know about OUD.

Based on Table 1 it looks like there may be some differences between cases and controls, but no significance test is performed. Also, how were controls identified? Did they have any previous opioid use? For all subjects, were opioids present in toxicology reports or in the medical record at time of death? Information about the brain samples themselves are lacking. For example, post-mortem interval, brain pH, and RINs are not reported.

The study is performed in all males, which is listed as a limitation in the discussion. This should be made more apparent by mentioning this in the abstract and introduction.

There are quite a number of significant individual findings for the sample size. This suggests there may potentially be an issue with test statistic inflation or many false positives. Can the authors report inflation measures, such as lambda, or provide QQ-plots to assess if this is a possibility?

Additional confidence that these findings are not false positives may be gained by a validation and replication effort.

On page 16, please clarify why a subtraction is done to obtain the hydroxy methylation levels as this may not be apparent to readers not familiar with technique.

For the GWAS enrichment analysis, it is unclear what is being tested. Can a sentence or two of additional description be added? Also, there are published OUD and opioid use GWAS studies. Were any of the genes identified in these studies found among the significant results?

All of the OUD cases, and some controls, had PTSD. While this was acknowledged as a limitation and PTSD was included as a covariate in all analyses, it would be nice to see a confirmation that previous PTSD-associated methylation sites were not among the significant findings.

Given the attention that has been placed on OPRM1 in the opioid genetics literature, I feel readers are going to expect an explicit statement or section on the findings in this gene.

Minor:

-Figure ordering does not match order presented in main text. Also, labels on sub-figures (e.g., A, B, C, etc.) do not match what is described in main text.

- Some text in Figures 4 & 5 is difficult to read. Also, figure descriptions could use more detail to help readers not familiar with these analyses.

- On page 5, the statement beginning with "Considering the gene nearest to each differential 5mC or 5hmC in OUD . . ." is confusing and needs to make clear that there were 56 overlapping genes between 5mC and 5hmC.

- On page 7, sentence beginning with "GO analysis revealed that the shared genes were associated with axon guidance . . ." should be made clear if these are genes that overlapped with the heroin methylation study or those that overlapped between 5mC and 5hmC.

- Figures 6A & 6B are hard to interpret. How should these figures be read? What is on the y-axis?

- In supplement Table 6, it would be easier to see where the overlap is among the four gene sets if genes common across sets were listed on the same line.

- For the GO analysis, why were sets with zero overlap with differential 5mC and 5hmC tested?

- How were significant findings mapped to genes for the GO and pathway analyses? Did they have to fall in the gene boundary? A certain distance from the transcription start site?

- How were cis and trans defined for the gene expression correlation analysis?

- Data availability statement only pertains to summary level information. Will the actual data be made available?

Reviewer #3 (Remarks to the Author):

This study by Rompala et al investigated DNA methylation and DNA hydroxymethylation in neuronal nuclei of the orbitofrontal cortex of 12 individuals with opioid use disorder and 26 controls. They carried out various analyses and main findings are enrichment of neuronal function in 5hmC methylation. In addition, 397 5mC CpG and 1740 5hm CpG sites were differentially methylated between cases and controls. Several downstream analyses were carried out.

This paper is of interest to the field as it adds important assessment of hydroxy methylation in neuronal postmortem brain tissue. Weaknesses include the relatively small sample size and no biological validation or replications. Only male participants were included which is a major weakness.

Comments:

Please provide some more clinical information if available. How long was opioid use ongoing, was it IV use or prescription opioid use? Why were only males included in the study? How were co-morbidities accounted for?

The authors state they found for 5mC 397 differential CpGs (357 genes) and for 5hmC and 1,740 differential CpGs (1,453 genes); however, it is unclear how correction for multiple testing was performed and what their statistical threshold was. Please provide additional information in the main manuscript. (I saw FDR corrected p values in the supplement, perhaps a more stringent threshold would be more appropriate). Consider moving the top 10 targets into a table to the main manuscript.

The finding of the Hemoglobin Subunit Beta (HBB) gene being associated with OUD is interesting, and not surprising as the authors write this might be hypoxia related, possibly due to overdose death due to respiratory depression. This brings up an important issue with the study, namely to what degree are the findings due to underlying opioid use (presumably chronic use) and to what degree are they due to hypoxia/postmortem changes unrelated to opioids. This could be discussed in more details and perhaps animal models might help to disentangle this issue.

March 3rd, 2023

Dear reviewers,

We appreciate the positive comments on our manuscript NCOMMS-22-38482, highlighting its many strengths, and the valuable suggestions to improve it further. We have addressed your comments and revised the manuscript accordingly.

The following is a point-by-point summary of these changes as well as responses to concerns and suggestions. Revisions in the main text are highlighted in yellow.

Reviewer #1:

Major: There are substantial differences in the rates of several covariates, most notable in smoking (10/12 vs 13/26). Moreover, all OUD cases are also PTSD cases. While it is worth noting that the most important differences are included as covariates, which does reduce some concern, there are some issues with the presentation. First, given the small sample size, Fisher's exact test should be used to compare cases vs controls on demographic, psych characteristics. A cursory calculation shows that these groups differ significantly on PTSD and at a trend level on smoking. Second, the section in the text describing these covariates reports only the counts for the case group but not the control group, which is somewhat confusing.

We thank the reviewer for the suggestion. We updated Table 1 and the main text by including the demographic characteristics' comparison between the OUD and non-OUD groups. As suggested, we used Fisher's exact test for the discrete variables and T-statistic for the continuous variables. Information on the description of the covariates for both OUD and non-OUD groups is included in Table 1.

Results

“There was no significant difference in the age of death between OUD and non-OUD groups (p-value=0.15), and smoking (p-value= 0.08). However, we did observe a significance for post-traumatic stress disorder (PTSD; p-value=0.003), and major depressive disorder (MDD) (p-value=0.009)”

Major: There is no discussion on drug use history or on the availability of toxicology screens for the subjects or the causes of death. There are obvious questions one would ask about the cases and controls. Were the subjects intoxicated at the time of death? Did the cases have a history of OUD or were they actively using opioids? Do the cases and controls differ on the cause of death (e.g., overdose, accident)? This information would help dramatically with interpretation of most of the results but most notable the drug interaction results.

This is a very important point. We have now added information on the cause of death, including toxicology-related information at the time of death. Regarding the cause of death, all individuals in the OUD group died from alcohol and/or drug intoxication, compared to n=3 in the non-OUD group. Individuals in the OUD group that died from alcohol and/or drug intoxication varied in the type of drug they died from, these being mostly opioids and alcohol. This information has been included in the Methods and the Discussion of the revised manuscript.

Methods

“Postmortem human brain specimens were obtained from the National Post-Traumatic Stress Disorder (PTSD) Brain Bank⁵⁵ (NPBB), a brain tissue repository at the U.S. Department of Veterans Affairs (VA). Informed consent was provided next-of-kin. Brain tissue samples were collected from the orbitofrontal cortex (OFC; Brodmann Area 11). The non-OUD group was defined as those individuals without a history of OUD diagnosis. The cause of death in the non-OUD group included natural causes (n=13, 34.21%), suicide, (n=5, 13.16%) accident causes (n=7, 18.42%) mainly by body injuries and gunshots, and alcohol and drug intoxication (n=3, 7.89%; including two opioid-related intoxications). All individuals in the OUD group have a history of OUD diagnosis and died from drug and/or alcohol intoxication, including, but not limited to, opioids. Table 1 shows the demographics and clinical characteristics of the study cohort. Clinical diagnoses were conducted using two approaches, antemortem assessment protocol for antemortem donors and postmortem diagnostic assessment for postmortem donors as described by Friedman and colleagues⁵⁵.” This study was approved by the Department of Veterans Affairs and Yale School of Medicine.

Discussion

“Our gene-drug interaction analyses with opioids showed a greater number of associations with differential 5hmC marks than with 5mC marks, suggesting a higher functional role of 5hmC in OUD. (...) While these associations may be driven by opioid intoxication at the time of death, our GWAS enrichment findings further support the interpretation that the identified epigenetic marks are functional, particularly 5hmC, in OUD. Future work in *in vivo* and *in vitro* models will confirm these findings and disentangle the potential effects of opioid overdose death from the direct role of 5mC and 5hmC play in OUD’s etiology. Toward that end, 5mC holds promise for pinpointing new targets for OUD treatment”

Minor: According to the manuscript, the enrichment analyses relied on the “nearest gene”. Was there a maximum distance where a CpG was deemed “not in a gene”?

We thank the reviewer for the question. The transcript ID annotation was automatically assigned to the closest transcript start site (TSS) using the Genomation R package, which considers any distance between the site and the TSS. Gene symbol was further assessed using the biomaRt R package.

To address the reviewer’s question, gene region location (i.e., gene body, upstream, downstream) was annotated using the UCSC genome browser, considering a maximum distance of 1500pb from the gene extremity (5’ and 3’ ends). This information has been added to Supplementary Tables 1 and 2.

Methods

“Annotation of the closest Ensembl transcript ID of the differential sites was conducted using Genomation⁶⁴ R package considering any distance between the site and the transcript start site (TSS). Annotation of gene symbol was conducted using the biomaRt⁶⁵ R package. Gene region location was annotated using the UCSC genome browser⁶⁶ and applying a threshold of 1500bp⁶⁷ (“Position to the Nearest gene ID” in **Supplementary Table 1** and **Supplementary Table 2**).”

Minor: I think it is important to examine the overlap with existing literature, but the authors looked at one study. They should provide a rationale justifying that one study. It is also important to address whether or not the direction (hypo vs hyper) is the same as in those studies versus just reporting overlap in differential methylation/hydroxymethylation?

We thank the reviewer for the suggestion, which can help increase the confidence of our findings.

We analyzed the overlap with the Kozlenkov et al. study to replicate our findings and assess its specificity to OUD. The Kozlenkov et al. study evaluated 5mC in the same brain region as in our study, the OFC. Further, they include individuals diagnosed with heroin abuse, with no history of alcoholism or other illicit drugs, so it was specific to heroin abuse.

There have been two additional recently published studies assessing 5mC in human postmortem brain tissue in opioid-related traits (Shu et al., 2021; Liu et al., 2021). These studies have examined the dorsolateral prefrontal cortex, a different brain region than our study; one of these studies focused on OUD (with many individuals reporting polydrug use), while the other one focused on acute opioid intoxication. Given the potential brain region specificity of 5mC patterns and the need to assess OUD-related specificity, we selected the Kozlenkov et al. study for replication purposes.

Following the reviewer’s suggestion, we expanded this query beyond the human postmortem brain, by also evaluating studies in human peripheral tissue, including blood (Montalvo-Ortiz et al., 2019; Lee et al., 2023), saliva (Sandoval-Sierra et al., 2023), and placenta (Borrelli et al., 2022; Radhakrishna et al., 2021). We have added these results in the Supplementary Table 8 of the revised manuscript and updated our Discussion accordingly.

Results

“We compared our findings with Kozlenkov et al.¹² that examined 5mC in the postmortem OFC of individuals with heroin abuse (N=87) using the Illumina 450K Infinium microarray. Very few differentially methylated CpGs identified in that study had sufficient read coverage to be examined in the present study (175 of 1391). None of these sites were differential 5mC or 5hmC CpGs in OUD cases, which might be attributable to technical differences, low coverage in these CpG sites, or demographic divergences between the studies. But when comparing genes with differential CpGs and using the same significance threshold as in Kozlenkov et al. ($p < 0.001$)¹², we found that 256 CpGs are commonly differentially methylated between opioid overdose¹² and OUD. The overlap increases to 327 CpGs when considering both 5mC and 5hmC (**Figure 2E**, **Supplementary Table 6**). GO analysis revealed that the overlapped genes with Kozlenkov et al. genes were associated with axon guidance (5mC+5hmC; odds ratio=3.08, FDR-adjusted p-value<0.001, **Supplementary Table 7**).

We also evaluated the overlap between our findings and the opioid epigenomic literature. We found an overlap of three genes (*SMARCA4*¹⁴, *JUN*¹⁴, *TAF3*¹³) when considering recent 5mC studies evaluating the DIPFC in opioid-related traits. Regarding 5mC studies in peripheral tissue, we found one overlapping gene from an EWAS study of opioid self-administration in patients who underwent dental surgery²², 127 overlapping genes when compared to two studies in the placenta evaluating neonatal opioid withdrawal

syndrome^{23,24}, and 6 from a recent study of opioid medication use²⁵ (**Supplementary Table 8**).”

Discussion

“Genome-wide differential CpG analysis identified 397 CpGs for 5mC and 1,740 CpGs for 5hmC associated with OUD. Many of these associations were previously found in other studies of opioid-related traits. For instance, 327 genes (127 with 5mC; **Supplementary Table 6**) were found in a prior 5mC study of postmortem OFC from individuals with heroin abuse who died from overdose¹². Similarly, two additional genes (*SMARCA4* and *JUN*) were implicated in recent 5mC studies in the dlPFC of opioid-related traits^{13, 14}, and identified as potential transcriptional regulators in a multi-omics study of OUD¹⁴. We also observed a gene-level overlap in 5mC studies evaluating peripheral tissues, including placenta^{23, 24}, saliva²², and blood²⁵. Most of the convergence was observed in 5mC studies of the placenta evaluating neonatal opioid withdrawal syndrome^{23, 24}. *OPRM1* is one of the most reported associated genes in the opioid literature. This gene was identified in our 5mC co-methylation analysis, specifically in the OUD-associated lightpink2 module. Taken together, these results demonstrate that our findings are consistent with the literature, adding confidence to previously reported associations while also revealing novel ones.”

Minor: The entire manuscript could use a proofread. There are issues with noun verb agreement, especially in the discussion, open parentheses.

We proofread the manuscript, following the reviewer’s suggestion.

Minor: The labels on figures are not always informative.

We thank the reviewer for the comment. We updated all the figure labels by adding more information.

“Figure 1. Analyzing the Neuronal Methylome and Hydroxymethylome in OUD. A) Experimental workflow. Fluorescence-Activated Nuclei Sorting was used to isolate neuronal nuclei from postmortem OFC. Nuclei were processed for genomic DNA, undergoing reduced representation oxidative bisulfite sequencing to examine 5mC and 5hmC at CpG-dense loci. (B) An average of 10x coverage was obtained for ~3.5 million CpG sites. C) 44% of CpG sites were located in promoter regions and 30% in intergenic regions. D-E) Neuronal 5mC occurs mainly in intergenic regions, while 5hmC occurs in introns, exons, and intergenic regions. F) Contrasting mean 5mC levels at neuronal (NeuN+) vs. non-neuronal (NeuN-) marker genes.

Figure 2. Differential 5mC and 5hmC marks of OUD. A) Distribution of differential sites into Chromosomal location of hyper- and hypomethylation for 5mC and 5hmC. B) Comparison between differential 5mC and 5hmC sites for CpG and gene, showing a gene overlap of 56 annotated genes. C) Distribution of differential CpGs across genomic loci showed the promoter region with the highest number of 5mC and 5hmC differential CpGs. D-E) Enrichment analysis for neuronal regions showed that 5mC and 5hmC differential CpGs were mostly enriched for neuronal H3K27ac. F) Comparison of our differential 5mC and 5hmC CpGs with differential 5mC markers detected in the OFC of heroin abuse cases (Kozlenkov et al.) showed an overlap of 129 genes. G) Gene ontology analysis for OUD-linked 5mC genes. H) Gene ontology analysis for OUD-linked 5hmC genes.

Figure 3. Drug interaction analysis of 5mC and 5hmC differential CpG sites with opioids. The plot shows the number of described drug interactions for each gene (left-axis; line weight indicates the number), highlighting its interaction with opioid-related drugs (right-axis). A) genes annotated with 5mC

ODU- associated differential CpG sites. B) genes annotated with 5hmC ODU-associated differential CpG sites.

Figure 4. Co-methylation of 5mC and 5hmC sites. A) 5mC-significant modules associated with ODU (correlation \geq |0.4|, p-value $<$ 0.05). B) 5hmC-significant modules associated with ODU (correlation \geq |0.4|, p-value $<$ 0.05). C) GO enrichment analysis for 5mC-significant modules (top 10 terms). D) GO enrichment for 5hmC-significant modules (top 10 terms).

Figure 5. Network of 5mC- and 5hmC-significant modules associated with ODU. The figure shows the PPI analyses for 9 modules detected with a significant association with ODU in 5mC (wheat1, ghostwhite, turquoise3, aquamarine1, turquoise1) and 5hmC (navajowhite3, orange4, rosybrown1, thistle1). The colors represent a different biological process in which the modules were enriched. In ODU-associated 5hmC modules, we observed a higher number of annotated genes involved in neurogenesis and regulation of neurogenesis.

Figure 6. Differential expression analysis of ODU. A) Volcano plot (threshold: most significant, 10^{-5}). *HBB* is identified as a differentially expressed gene associated with ODU. B) Venn diagram showing the overlap between differential 5mC CpG sites and differentially expressed genes, with an overlap of 64 genes. C) Venn diagram showing the overlap between differential 5hmC CpG sites and differentially expressed genes, with an overlap of 257 genes.”

Minor: The background literature is much more substantial than the one or two papers cited. I think it is important to include more of this to highlight that a substantial amount of work has been done on methylation but much less on hydroxymethylation.

We thank the reviewer for the suggestion. We expanded and updated the background literature in the Introduction and Discussion of our revised manuscript.

Introduction

“Multiple kinds of epigenetic modifications regulate gene expression that impacts behaviors relevant to opioid addiction¹⁰. We and others have evaluated the relationship between 5mC and opioid-related traits in human peripheral tissue and postmortem brain tissue. For example, an epigenome-wide association study (EWAS) in European women identified three differential 5mC sites associated with opioid dependence in peripheral blood¹¹. In the human postmortem brain, a study evaluating 5mC in individuals with heroin abuse that died from overdose reported differential 5mC of several gene classes, including those implicated in glutamate neurotransmission, axonogenesis, synaptic processes, and gene regulation¹². A more recent 5mC study in the dorsolateral prefrontal cortex (dlPFC) reported 13 differential CpG sites nominally associated with opioid intoxication with a suggestive significance of $p < 1.0 \times 10^{-5}$; with no differential CpG sites surviving multiple testing correction¹³. Another recent study in the dlPFC conducted an integrative analysis of epigenomic and transcriptomic data in the context of OUD, identifying potential regulatory genes associated with OUD-related expression patterns, and co-methylated modules associated with OUD and enriched for neurogenesis and neuronal development¹⁴.”

Discussion

“Genome-wide differential CpG analysis identified 397 CpGs for 5mC and 1,740 CpGs for 5hmC associated with OUD. Many of these associations were previously found in other studies of opioid-related traits. For instance, 327 genes (127 with 5mC; **Supplementary Table 6**) were found in a prior 5mC study of postmortem OFC from individuals with heroin abuse who died from overdose¹². Similarly, two additional genes (*SMARCA4* and *JUN*) were implicated in recent 5mC studies in the dlPFC of opioid-related traits^{13, 14}, and identified as potential transcriptional regulators in a multi-omics study of OUD¹⁴. We also observed a gene-level overlap in 5mC studies evaluating peripheral tissues, including placenta^{23, 24}, saliva²², and blood²⁵. Most of the convergence was observed in 5mC studies of the placenta evaluating neonatal opioid withdrawal syndrome^{23, 24}. *OPRM1* is one of the most reported associated genes in the opioid literature. This gene was identified in our 5mC co-methylation analysis, specifically in the OUD-associated lightpink2 module. Taken together, these results demonstrate that our findings are consistent with the literature, adding confidence to previously reported associations while also revealing novel ones.”

Minor: If there are not limitations on the words, the introduction should set the stage for this work more thoroughly. The authors should consider their target readership in editing this manuscript. It could be substantially more accessible.

We have updated the Introduction accordingly and considering the word limit.

Minor: The authors do not plan to share data other than summary statistics. This is problematic in the current data sharing environment.

We thank the reviewer for this comment. Acknowledging the importance of sharing data beyond summary statistics, raw data will be deposited in dbGAP for public access. This process is ongoing and we expect to complete it before the publication of our manuscript. We are currently waiting for the approval of our Data User Agreement at Yale.

Reviewer #2 (Remarks to the Author):

The overall goal of this study was to examine DNA methylation (5mC) and hydroxymethylation (5hmC) disturbances in neurons sorted from human postmortem brain samples of individuals with opioid use disorder (OUD) versus unaffected controls. The authors report identifying significant findings in both 5mC and 5hmC, correlation of sites with gene expression sequencing data from the same individuals and OUD-associated differentially expressed genes, and other results from additionally bioinformatic analyses. The manuscript is organized and well-written, and the findings are of importance to the field – though the importance of these findings is not always made apparent. Areas of concern are outlined below.

There are a handful of other studies examining DNA methylation and OUD or opioid use, however these studies and their findings are not currently discussed in the manuscript. Is there any overlap between the findings in this study and the previous studies? Perhaps the authors could perform a look-up replication of previous findings in their results?

We thank the reviewer for the suggestion. We have evaluated the overlap of our findings with previous studies and included them in the revised manuscript.

Results

“The overlap increases to 327 CpGs when considering both 5mC and 5hmC (**Figure 2F**, **Supplementary Table 6**). GO analysis revealed that the overlapped genes with Kozlenkov et al. were associated with axon guidance (5mC+5hmC; odds ratio=3.08, FDR-adjusted p-value<0.001, **Supplementary Table 7**). We also evaluated the overlap between our findings and the opioid epigenomic literature. We found an overlap of three genes (*SMARCA4*¹⁴, *JUN*¹⁴, *TAF3*¹³) when considering recent 5mC studies evaluating the dIPFC in opioid-related traits. Regarding 5mC studies in peripheral tissue, we found one overlapping gene from an EWAS study of opioid self-administration in patients who underwent dental surgery²², 127 overlapping genes when compared to two studies in the placenta evaluating neonatal opioid withdrawal syndrome^{23,24}, and 6 from a recent study of opioid medication use²⁵ (**Supplementary Table 8**).”

Discussion

“Genome-wide differential CpG analysis identified 397 CpGs for 5mC and 1,740 CpGs for 5hmC associated with OUD. Many of these associations were previously found in other studies of opioid-related traits. For instance, 327 genes (127 with 5mC; **Supplementary Table 6**) were found in a prior 5mC study of postmortem OFC from individuals with heroin abuse who died from overdose¹². Similarly, two additional genes (*SMARCA4* and *JUN*) were implicated in recent 5mC studies in the dIPFC of opioid-related traits^{13,14}, and identified as potential transcriptional regulators in a multi-omics study of OUD¹⁴. We also observed a gene-level overlap in 5mC studies evaluating peripheral tissues, including placenta^{23,24}, saliva²², and blood²⁵. Most of the convergence was observed in 5mC studies of the placenta evaluating neonatal opioid withdrawal syndrome^{23,24}. *OPRM1* is one of the most reported associated

genes in the opioid literature. This gene was identified in our 5mC co-methylation analysis, specifically in the OUD-associated lightpink2 module. Taken together, these results demonstrate that our findings are consistent with the literature, adding confidence to previously reported associations while also revealing novel ones.”

The discussion currently reads as a re-statement of all the results. While, some of this is needed given the number of analyses, it is hard to know which findings are really important, why they are important them and how to interpret them within the context of what we know about OUD.

We have revised the Discussion to emphasize the most important findings and their interpretation in the context of OUD’s literature. For example, we highlight the discussion and interpretation of our findings in the overlap with the existing literature, the OUD GWAS enrichment, as well as the gene-drug interaction findings.

Based on Table 1 it looks like there may be some differences between cases and controls, but no significance test is performed. Also, how were controls identified? Did they have any previous opioid use? For all subjects, were opioids present in toxicology reports or in the medical record at time of death? Information about the brain samples themselves are lacking. For example, post-mortem interval, brain pH, and RINs are not reported.

We conducted a comparison of demographic characteristics between the OUD and non-OUD groups, now included in Table 1. We used Fisher’s exact test for the discrete variables and T-statistic for the continuous variables. The non-OUD group was defined as those individuals without a diagnosis of OUD. While all individuals in the OUD group died from drug and/or alcohol intoxication, including opioid-related drugs (e.g., heroin, fentanyl, and methadone), death by drug intoxication is also observed in the non-OUD group (n=3, two of which were from opioid-related drugs). This toxicology information was added to the Methods section. Post-mortem interval (PMI) is included in Table 1, however, pH information is not available. The RIN information is included in the Methods section of the transcriptomic analysis.

Methods

“The non-OUD group was defined as those individuals without a history of OUD diagnosis. The cause of death in the non-OUD group included natural causes (n=13, 34.21%), suicide, (n=5, 13.16%) accident causes (n=7, 18.42%) mainly by body injuries and gunshots, and alcohol and drug intoxication (n=3, 7.89%, including two opioid-related intoxications). All individuals in the OUD group have a history of OUD diagnosis and died from drug and/or alcohol intoxication, including, but not limited to, opioids.”

Results

“RIN values for the non-OUD and OUD groups were similar (mean non-OUD = 7.64±1.14 and mean OUD = 8.38±0.50.”

The study is performed in all males, which is listed as a limitation in the discussion. This should be made more apparent by mentioning this in the abstract and introduction.

As suggested, this information is now included in the Abstract and Introduction.

Abstract

“Here, we conducted a multi-omics profiling of the orbitofrontal cortex (OFC) of OUD in a male cohort,

integrating neuronal-specific 5mC and 5hmC as well as within-subject correlations with gene expression profiles from human postmortem samples (OUD=12; non-OUD=26).”

Introduction

“Here, we conducted the first parallel 5mC and 5hmC profiling of OUD in neuronal nuclei from human postmortem OFC of a male cohort.”

There are quite a number of significant individual findings for the sample size. This suggests there may potentially be an issue with test statistic inflation or many false positives. Can the authors report inflation measures, such as lambda, or provide QQ-plots to assess if this is a possibility?

We thank the reviewer for this very important point. Inflation measures (i.e., λ values) and QQ-plots are reported in the revised manuscript and the related figure is added as **Supplementary Figure 3**. There is no evidence of inflation in our results.

Results

“ λ values for 5mC and 5hmC differential analyses were 1.07 and 0.92, respectively. QQ-plots are included in **Supplementary Figure 3**.”

Additional confidence that these findings are not false positives may be gained by a validation and replication effort.

We thank the reviewer for this suggestion. In addition to comparing our results to Kozlenkov et al., we have evaluated our findings with two recently published 5mC studies in human postmortem brain (i.e., dIPFC) and even expanded our query to 5mC studies examining other tissues to reinforce the confidence of our findings.

Results

“The overlap increases to 327 CpGs when considering both 5mC and 5hmC (**Figure 2F**, **Supplementary Table 6**). GO analysis revealed that the overlapped genes with Kozlenkov et al. genes were associated with axon guidance (5mC+5hmC; odds ratio=3.08, FDR-adjusted p-value<0.001, **Supplementary Table 7**). We also evaluated the overlap between our findings and the opioid epigenomic literature. We found an overlap of three genes (*SMARCA4*¹⁴, *JUN*¹⁴, *TAF3*¹³) when considering recent 5mC studies evaluating the dIPFC in opioid-related traits. Regarding 5mC studies in peripheral tissue, we found one overlapping gene from an EWAS study of opioid self-administration in patients who underwent dental surgery²², 127 overlapping genes when compared to two studies in the placenta evaluating neonatal opioid withdrawal syndrome^{23,24}, and 6 from a recent study of opioid medication use²⁵ (**Supplementary Table 8**).”

Discussion

“Genome-wide differential CpG analysis identified 397 CpGs for 5mC and 1,740 CpGs for 5hmC associated with OUD. Many of these associations were previously found in other studies of opioid-related traits. For instance, 327 genes (127 with 5mC; **Supplementary Table 6**) were found in a prior 5mC study of postmortem OFC from individuals with heroin abuse who died from overdose¹². Similarly, two additional genes (*SMARCA4* and *JUN*) were implicated in recent 5mC studies in the dIPFC of opioid-related traits^{13, 14}, and identified as potential transcriptional regulators in a multi-omics study of OUD¹⁴. We also observed a gene-level overlap in 5mC studies evaluating peripheral tissues, including placenta^{23, 24}, saliva²², and blood²⁵. Most of the convergence was observed in 5mC studies of the placenta evaluating neonatal opioid withdrawal syndrome^{23, 24}. *OPRM1* is one of the most reported associated

genes in the opioid literature. This gene was identified in our 5mC co-methylation analysis, specifically in the OUD-associated lightpink2 module. Taken together, these results demonstrate that our findings are consistent with the literature, adding confidence to previously reported associations while also revealing novel ones.”

On page 16, please clarify why a subtraction is done to obtain the hydroxy methylation levels as this may not be apparent to readers not familiar with technique.

Following the reviewer’s suggestion, we included more information in the Methods section describing how 5hmC is calculated, resulting from the subtraction of the BS and oxBS libraries.

Methods

“Raw data were processed by Weill Cornell Epigenomics Core (New York, NY). Bismark bisulfite read mapper⁵⁶ was used to map sequencing reads to the CRCh38 human genome and detect bisulfite treatment of converted and unconverted cytosines. Two libraries of each sample were performed for sequencing: 1. Bisulfite sequencing (BS), which changes unmethylated and unhydroxymethylated cytosines into thymine; and 2. Oxidative BS (oxBS), which allows the conversion of hydroxymethylated cytosines into thymine. Beta values (% unconverted cytosines) are calculated for 5hmC as the difference in methylation at each CpG site between BS and oxBS libraries (e.g., if $\beta=100\%$ in the BS library and $\beta_{5mC}=60\%$ in the oxBS library, then $5hmC=40\%$). Analyzed CpGs were filtered for a minimum of 10x coverage in all subjects and were normalized for coverage variability.”

For the GWAS enrichment analysis, it is unclear what is being tested. Can a sentence or two of additional description be added? Also, there are published OUD and opioid use GWAS studies. Were any of the genes identified in these studies found among the significant results?

Following the reviewer’s comments, we have added a more detailed description of the GWAS enrichment methodology and findings. Since the GWAS enrichment analysis was performed using FUMA (Watanabe et al., 2017), which does not include the recent OUD GWAS, we performed a new GWAS enrichment analysis using the data from a recently published OUD GWAS that identified 19 independent risk loci (Deak et al., 2022).

Results

“We further tested the OUD GWAS enrichment of the differential 5mC and 5hmC genes using data from a recent OUD GWAS in individuals of African and European ancestry³¹. For 5mC, we found 5 overlapping genes (Supplementary Table 8) but did not observe a significant enrichment (OR=1.4371, p-value=0.2758). However, for 5hmC differential genes, we observed a significant enrichment (OR=1.8757, p-value =0.0028) with 27 overlapped genes (Supplementary Table 8).”

Methods

“The analysis was performed using the online website FUMA⁷⁴. First, the Ensembl IDs were transformed to Entrez IDs using David⁷⁵ and submitted to FUMA’s GENE2Function web tool. Once the enrichment analysis was generated, the results were merged with the domains and traits reported by the GWAS Atlas using in-house scripts. We further conducted the GWAS enrichment for OUD using data from a recently published GWAS in European and African populations³¹. For this, we used Fisher’s exact test and a similar background to the Metascape platform (30,182 genes)⁷⁴. Significance was defined as $FDR < 0.05$.”

All of the OUD cases, and some non-OUD, had PTSD. While this was acknowledged as a limitation and

PTSD was included as a covariate in all analyses, it would be nice to see a confirmation that previous PTSD-associated methylation sites were not among the significant findings.

This is a very good point, and we thank the reviewer for the suggestion. We have compared our findings with previously reported PTSD-associated differential 5mC genes. For 5mC, from the 397 associations identified in our study, only one (*HOOK2*) has been previously reported in the 5mC PTSD literature, with the same direction effect. In terms of 5hmC, from the 1,740 associations identified here, only 6 were previously reported in 5mC studies of PTSD (four of these observed in opposite directions; *COLIA2*, *DUSP22*, *AHRR*, *NRG1*, *ZBED4*, *GTF2IRD1*). We have included this information in the Discussion and Supplementary Tables 1 and 2.

Results

“We further evaluated the gene-level overlap of our findings and the previously reported PTSD 5mC associations to verify PTSD-related confounder effects (“PTSD-associated genes” in **Supplementary Tables 1 and 2**). We found very little overlap considering both 5mC and 5hmC differential sites, suggesting that our findings are not driven by PTSD comorbidity.”

Given the attention that has been placed on OPRM1 in the opioid genetics literature, I feel readers are going to expect an explicit statement or section on the findings in this gene.

We thank the reviewer for this important suggestion. In our findings, although we did not observe the *OPRM1* in the differential CpG analysis, we did identify this gene in the co-methylation analysis, specifically in the lightpink2 module. We updated the results and discussion accordingly.

Results

“In our module-based analysis, we found *OPRM1* in the lightpink2 module, a gene commonly reported in the opioid literature.”

Discussion

“*OPRM1* is one of the most reported associated genes in the opioid literature. This gene was identified in our 5mC co-methylation analysis, specifically in the OUD-associated lightpink2 module. Taken together, these results demonstrate that our findings are consistent with the literature, adding confidence to previously reported associations while also revealing novel ones.”

Minor:

-Figure ordering does not match order presented in main text. Also, labels on sub-figured (e.g., A, B, C, etc.) do not match what is described in main text.

We thank the reviewer for the careful reading of our manuscript. We updated all the figure's order and labels to make sure that the figure ordering matches the order presented in the main text.

- Some text in Figures 4 & 5 is difficult to read. Also, figure descriptions could use more detail to help readers not familiar with these analyses.

We thank the reviewer for noticing this. We updated all the figures improving the resolution and increasing the font size of the text. We also updated the figures' description (see below).

“**Figure 1. Analyzing the Neuronal Methylome and Hydroxymethylome in OUD.** A) Experimental

workflow. Fluorescence-Activated Nuclei Sorting was used to isolate neuronal nuclei from postmortem OFC. Nuclei were processed for genomic DNA, undergoing reduced representation oxidative bisulfite sequencing to examine 5mC and 5hmC at CpG-dense loci. (B) An average of 10x coverage was obtained for ~3.5 million CpG sites. (C) 44% of CpG sites were located in promoter regions and 30% in intergenic regions. (D-E) Neuronal 5mC occurs mainly in intergenic regions, while 5hmC occurs in introns, exons, and intergenic regions. (F) Contrasting mean 5mC levels at neuronal (NeuN+) vs. non-neuronal (NeuN-) marker genes.

Figure 2. Differential 5mC and 5hmC marks of OUD. A) Distribution of differential sites into Chromosomal location of hyper- and hypomethylation for 5mC and 5hmC. B) Comparison between differential 5mC and 5hmC sites for CpG and gene, showing a gene overlap of 56 annotated genes. C) Distribution of differential CpGs across genomic loci showed the promoter region with the highest number of 5mC and 5hmC differential CpGs. D-E) Enrichment analysis for neuronal regions showed that 5mC and 5hmC differential CpGs were mostly enriched for neuronal H3K27ac. F) Comparison of our differential 5mC and 5hmC CpGs with differential 5mC markers detected in the OFC of heroin abuse cases (Kozlenkov et al.) showed an overlap of 129 genes. G) Gene ontology analysis for OUD-linked 5mC genes. H) Gene ontology analysis for OUD-linked 5hmC genes.

Figure 3. Drug interaction analysis of 5mC and 5hmC differential CpG sites with opioids. The plot shows the number of described drug interactions for each gene (left-axis; line weight indicates the number), highlighting its interaction with opioid-related drugs (right-axis). A) genes annotated with 5mC OUD-associated differential CpG sites. B) genes annotated with 5hmC OUD-associated differential CpG sites.

Figure 4. Co-methylation of 5mC and 5hmC sites. A) 5mC-significant modules associated with OUD (correlation \geq |0.4|, p-value $<$ 0.05). B) 5hmC-significant modules associated with OUD (correlation \geq |0.4|, p-value $<$ 0.05). C) GO enrichment analysis for 5mC-significant modules (top 10 terms). D) GO enrichment for 5hmC-significant modules (top 10 terms).

Figure 5. Network of 5mC- and 5hmC-significant modules associated with OUD. The figure shows the PPI analyses for 9 modules detected with a significant association with OUD in 5mC (wheat1, ghostwhite, turquoise3, aquamarine1, turquoise1) and 5hmC (navajowhite3, orange4, rosybrown1, thistle1). The colors represent a different biological process in which the modules were enriched. In OUD-associated 5hmC modules, we observed a higher number of annotated genes involved in neurogenesis and regulation of neurogenesis.

Figure 6. Differential expression analysis of OUD. A) Volcano plot (threshold: most significant, 10^{-5}). *HBB* is identified as a differentially expressed gene associated with OUD. B) Venn diagram showing the overlap between differential 5mC CpG sites and differentially expressed genes, with an overlap of 64 genes. C) Venn diagram showing the overlap between differential 5hmC CpG sites and differentially expressed genes, with an overlap of 257 genes.

- On page 5, the statement beginning with “Considering the gene nearest to each differential 5mC or 5hmC in OUD . . . “ is confusing and needs to make clear that there were 56 overlapping genes between 5mC and 5hmC.

We have updated this sentence following the reviewer’s suggestion.

Results

“There was no overlap between differential 5mC and 5hmC marks. However, at the gene level, we observed 56 overlapping genes between 5mC and 5hmC differential marks (15.4% of 5mC-linked genes and 3.5% of 5hmC-linked genes) (Figures 2A and 2B).”

- On page 7, sentence beginning with “GO analysis revealed that the shared genes were associated with axon guidance . . .” should be made clear if these are genes that overlapped with the heroin methylation study or those that overlapped between 5mC and 5hmC.

We thank the reviewer for the suggestion. We updated the text to clarify the reviewer’s question.

Results

“GO analysis revealed that the overlapped genes with Kozlenkov et al. were associated with axon guidance (5mC+5hmC; odds ratio=3.08, FDR-adjusted p-value<0.001, Supplementary Table 7).”

- Figures 6A & 6B are hard to interpret. How should these figures be read? What is on the y-axis?

Figure 6 is now Figure 3 in the revised manuscript. We updated the figure and the labels to make them clearer (see below). The y-axis indicates the number of described drug interactions in the DGIdb database (Freshour, 2021).

“Figure 3. Drug interaction analysis of 5mC and 5hmC differential CpG sites with opioids. The plot shows the number of described drug interactions for each gene (left-axis; line weight indicates the number), highlighting its interaction with opioids (right-axis). A) genes annotated with OUD-associated 5mC differential CpG sites. B) genes annotated with OUD-associated differential 5hmC CpG sites.”

- In supplement Table 6, it would be easier to see where the overlap is among the four gene sets if genes common across sets were listed on the same line.

We thank the reviewer for this suggestion. We updated Supplementary Table 6 to keep the genes in each analysis in the same row.

- For the GO analysis, why were sets with zero overlap with differential 5mC and 5hmC tested?

GO analysis was conducted for the overlap between 5mC, 5hmC, and Kozlenkov et al’s results.

Results

“The overlap increases to 327 CpGs when considering both 5mC and 5hmC (Figure 2F, Supplementary Table 6). GO analysis revealed that the overlapped genes with Kozlenkov et al. were associated with axon guidance (5mC+5hmC; odds ratio=3.08, FDR-adjusted p-value<0.001, Supplementary Table 7).”

- How were significant findings mapped to genes for the GO and pathway analyses? Did they have to fall in the gene boundary? A certain distance from the transcription start site?

We thank the reviewer for the question. The transcript ID annotation was automatically assigned to the closest transcript start site (TSS) using the Genomation R package, considering any distance between the site and TSS. Gene symbol was further assessed using the biomaRt R package.

To address the reviewer's question, gene region location (i.e., gene body, upstream, downstream) was annotated using the UCSC genome browser, considering a maximum distance of 1500pb from the gene extremity (5' and 3' ends). This information has been added to Supplementary Tables 1 and 2. Considering that less than 5.1% of the sites were annotated in the intergenic region, we used all the annotated gene symbols for the subsequent analyses.

Methods

“Annotation of the closest Ensembl transcript ID of the differential sites was conducted using Genomation⁶⁴ R package considering any distance between the site and the transcript start site (TSS). Annotation of gene symbol was conducted using the biomaRt⁶⁵ R package. A complimentary annotation was conducted using the UCSC genome browser⁶⁶ to identify gene region location applying a threshold of 1500bp⁶⁷ (“Position to the Nearest gene ID” in **Supplementary Table 1** and **Supplementary Table 2**).”

How were cis and trans defined for the gene expression correlation analysis?

To define cis, we used the default parameters from the *Matrix_eQTL_main* function (Shabalín et al., 2012), that is, any site from the 1e-6 bp from the starting or ending of the annotated gene. For trans, we used everything outside that threshold. This description was included in the Methods section of the revised manuscript.

Methods

“Pearson correlation between mRNA analysis and 5mC/5hmC data was calculated using the MatrixEQTL package⁷³. Correlation analysis was performed in *cis* and *trans*. To define a *cis*, we used the default parameters from the *Matrix_eQTL_main* function, that is, any site from the 1e-6 bp from the starting or ending of the annotated gene. For the *trans*, we used everything outside that threshold.”

- Data availability statement only pertains to summary level information. Will the actual data be made available?

We thank the reviewer for this comment. Raw data will be available in dbGAP for public access. This process is ongoing and we expect to complete it before the publication of our manuscript. We are currently waiting for the approval of our Data User Agreement at Yale.

Reviewer #3 (Remarks to the Author):

This study by Rompala et al investigated DNA methylation and DNA hydroxymethylation in neuronal nuclei of the orbitofrontal cortex of 12 individuals with opioid use disorder and 26 controls. They carried out various analyses and main findings are enrichment of neuronal function in 5hmC methylation. In addition, 397 5mC CpG and 1740 5hm CpG sites were differentially methylated between cases and controls. Several downstream analyses were carried out.

This paper is of interest to the field as it adds important assessment of hydroxy methylation in neuronal postmortem brain tissue. Weaknesses include the relatively small sample size and no biological validation or replications. Only male participants were included which is a major weakness.

Comments:

Please provide some more clinical information if available. How long was opioid use ongoing, was it IV

use or prescription opioid use? Why were only males included in the study? How were co-morbidities accounted for?

We thank the reviewer for this important point. We have added additional clinical information about our cohort including the cause of death and toxicology information at the time of death in the Methods section. Unfortunately, information on opioid use history, route of administration, or medication history is not available.

Since we were limited by sample size, only males were included to reduce the heterogeneity in the cohort. However, ongoing efforts include increasing the sample size by adding female samples. Regarding the co-morbidities, we included PTSD and smoking as covariates. PTSD comorbidity was also addressed by doing a gene-level look up of our findings with previous 5mC PTSD studies where we found very little overlap. Additional comorbidities will be evaluated in better-powered samples.

Methods

“ Postmortem human brain specimens were obtained from the National Post-Traumatic Stress Disorder (PTSD) Brain Bank⁵⁵ (NPBB), a brain tissue repository at the U.S. Department of Veterans Affairs (VA). Informed consent was provided next-of-kin. Brain tissue samples were collected from the orbitofrontal cortex (OFC; Brodmann Area 11). The non-ODU group was defined as those individuals without a history of OUD diagnosis. The cause of death in the non-ODU group included natural causes (n=13, 34.21%), suicide, (n=5, 13.16%) accident causes (n=7, 18.42%) mainly by body injuries and gunshots, and alcohol and drug intoxication (n=3, 7.89%, including two opioid-related intoxications). All individuals in the OUD group have a history of OUD diagnosis and died from drug and/or alcohol intoxication, including, but not limited to, opioids.

Table 1 shows the demographics and clinical characteristics of the study cohort. Clinical diagnoses were conducted using two approaches, antemortem assessment protocol for antemortem donors and postmortem diagnostic assessment for postmortem donors as described by Friedman and colleagues⁵⁵. This study was approved by the Department of Veterans Affairs and Yale School of Medicine.”

The authors state they found for 5mC 397 differential CpGs (357 genes) and for 5hmC and 1,740 differential CpGs (1,453 genes); however, it is unclear how correction for multiple testing was performed and what their statistical threshold was. Please provide additional information in the main manuscript. (I saw FDR corrected p values in the supplement, perhaps a more stringent threshold would be more appropriate). Consider moving the top 10 targets into a table to the main manuscript.

We thank the reviewer for these suggestions. We updated the Methods to include how the q-value (adjusted P-value) was calculated and the threshold used.

Methods

“Differential analysis of 5mC and 5hmC was performed at single-CpG resolution with the methylkit R package⁶⁰, using logistic regression with correction for overdispersion and chi-squared significance testing and benchmarked for the best balance of sensitivity and specificity⁶¹. P-value was calculated in the logistic regression and adjusted to q-value using Sliding Linear Model (SLIM)⁶², which uses simulated data accounting for data structure and hypothesis. Q-value thresholds were applied at 0.05. We also report Bonferroni-adjusted findings, included in **Supplementary Tables 1 and 2** to evaluate our findings using a more stringent threshold (5mC: p=2.71E-8; 5hmC: p=3.02E-8).”

Further, following the reviewer's suggestion, we included the top differential CpG 5mC and 5hmC sites associated with OUD in Table 2 of the main text.

Table 2: Top 10 genome-wide significant OUD-associated 5mC and 5hmC differential CpG sites. The table includes the chromosomal location, q-value statistics, effect size, annotated gene using nearest Gene ID, and position to the nearest gene ID (i.e., downstream, upstream, or in the gene body, considering a maximum TSS distance of 1500bp).

5mC					
Chr	Start	q-value	Δ 5mC	Nearest Gene ID	Position to the Nearest gene ID
chr19	54096075	1.13E-289	-0.40	OSCAR	Gene_body
chr7	108569763	4.00E-240	0.07	THAP5	Upstream
chr11	128891189	3.19E-185	-0.21	KCNJ5	Downstream
chr14	99480992	4.06E-183	0.22	SETD3	Upstream
chr19	56120943	2.36E-121	-0.21	ZNF787	Gene_body
chr14	105526298	8.33E-86	0.12	TMEM121	Downstream
chr11	3165259	4.65E-66	0.52	OSBPL5	Gene_body
chr6	12008931	1.95E-54	-0.26	HIVEP1	Gene_body
chrX	9465553	6.36E-53	-0.19	TBLIX	Gene_body
chr9	126805077	6.36E-53	-0.37	ZBTB43	Gene_body
5hmC					
Chr	Start	q-value	Δ 5hmC	Gene Symbol	Position to the Nearest gene ID
chr1	109042116	1.05E-306	-0.22	WDR47	Upstream
chr16	67416523	2.02E-306	-0.15	ZDHHC1	Gene_body
chr6	163413930	2.72E-305	0.03	CAHM	Gene_body
chr17	27793835	3.12E-291	-0.21	NOS2	Gene_body
chr5	181190518	3.74E-291	-0.21	LINC01962	Gene_body
chr19	639763	6.76E-291	0.40	FGF22	Downstream
chr4	37891265	2.85E-290	-0.31	TBC1D1	Gene_body
chr9	120928860	5.78E-287	0.08	TRAF1	Gene_body
chr5	149960747	6.32E-260	-0.22	SLC26A2	Downstream
chr2	96760842	6.92E-260	-0.19	CNNM4	Downstream

The finding of the Hemoglobin Subunit Beta (HBB) gene being associated with OUD is interesting, and not surprising as the authors write this might be hypoxia related, possibly due to overdose death due to respiratory depression. This brings up an important issue with the study, namely to what degree are the findings due to underlying opioid use (presumably chronic use) and to what degree are they due to hypoxia/postmortem changes unrelated to opioids. This could be discussed in more details and perhaps animal models might help to disentangle this issue.

We thank the reviewer for this very important comment. We would like to highlight that the *HBB* finding was observed in the transcriptomic analysis from bulk tissue, but not in our 5mC and 5hmC analysis from neuronal nuclei, which is the main focus of our paper. Also, we note that not all the individuals in the OUD group died from opioid overdose and that there are individuals in the non-OUD group that also died from opioid intoxication. Still, we agree with the reviewer that this is an important confounder to consider in our analysis when evaluating the validity of our transcriptomic findings.

We updated our discussion by expanding on the role of this gene across multiple domains, including hypoxia, but also immune response and neurodegeneration, processes that could be directly or indirectly related to OUD. We also discussed the potential effects of opioid intoxication at the time of death as a limitation and proposed the use of model organisms (including animal models) to help address this limitation.

We agree with the reviewer that animal models could be very informative in the interpretation of findings from human postmortem brain data. However, this work, at least at the genome-wide scale, is still emerging. A recent preprint from Dr. Eric Nestler's group at Mount Sinai, an expert in this field, reported the first comprehensive transcriptomic study of an animal model of heroin abuse evaluating different stages of drug addiction and assessing 8 different brain regions that are part of the addiction circuitry. (Browne et al., 2023). However, summary-level data is not available yet, which limited our ability to compare our findings with animal data. Further, no similar studies have been done evaluating 5mC and/or 5hmC in animal models of OUD.

Alternatively, we have evaluated our findings with previously published 5mC work in human postmortem brain and peripheral tissue and revised our manuscript accordingly. Last, but not least, we have also performed an OUD GWAS enrichment analysis and showed gene-level overlap for both 5mC and 5hmC as well as a significant enrichment for 5hmC. These GWAS enrichment results may support causal directionality in the context of OUD. Taken together, these analyses increase the confidence of our findings in the context of OUD.

Discussion

“To better understand the impact of 5mC and 5hmC alterations on gene expression, we conducted a differential gene expression analysis followed by integration with 5mC and 5hmC data. Differential analysis of RNA sequencing data from bulk tissue revealed an association of the *HBB* gene with OUD. *HBB* encodes a component of hemoglobin, the oxygen-carrying protein that also plays a role in hypoxia and immune response. *HBB* has been implicated in neurodegenerative processes of Parkinson's⁴², and Alzheimer's⁴³, and in opioid use^{44, 45}. Future studies are needed to confirm the relationship between *HBB* and OUD and address potential hypoxia-related effects in our OUD transcriptomic findings in human postmortem brain.”

(...)

“This study evaluated 5mC and 5hmC at CpG sites; future studies may evaluate the role of 5mC and 5hmC in non-CpG sites in the context of OUD. Non-CpG sites have been suggested to play a role in the development of neuropsychiatric diseases⁵⁴. Given the enrichment of differential 5mC and 5hmC genes and modules in GWAS studies, potential methylation/hydroxymethylation quantitative trait loci could be evaluated to assess whether differential 5mC/5hmC is influenced by genetic background.

Though the enrichment of GWAS signals may be indicative of potential causal effects of the OUD-associated 5mC/5hmC marks, research in model organisms (e.g., animal models, human induced pluripotent stem cells, brain organoids, etc.) may help discern whether the epigenetic marks are the cause or consequence of OUD and disentangle the effects of opioid overdose at the time of death from those involved in OUD.”

References:

- Kozlenkov A, et al. DNA Methylation Profiling of Human Prefrontal Cortex Neurons in Heroin Users Shows Significant Difference between Genomic Contexts of Hyper- and Hypomethylation and a Younger Epigenetic Age. *Genes (Basel)* **8**, (2017).
- Shu C, et al. Epigenome-wide study of brain DNA methylation following acute opioid intoxication. *Drug Alcohol Depend* **221**, 108658 (2021).
- Liu A, et al. Genome-Wide Correlation of DNA Methylation and Gene Expression in Postmortem Brain Tissues of Opioid Use Disorder Patients. *Int J Neuropsychopharmacol* **24**, 879-891 (2021).
- Montalvo-Ortiz JL, Cheng Z, Kranzler HR, Zhang H, Gelernter J. Genomewide Study of Epigenetic Biomarkers of Opioid Dependence in European- American Women. *Sci Rep* **9**, 4660 (2019).
- Lee M, et al. Opioid medication use and blood DNA methylation: epigenome-wide association meta-analysis. *Epigenomics* (2023).
- Sandoval-Sierra JV, Salgado Garcia FI, Brooks JH, Derefinko KJ, Mozhui K. Effect of short-term prescription opioids on DNA methylation of the OPRM1 promoter. *Clin Epigenetics* **12**, 76 (2020).
- Borrelli KN, et al. Effect of Prenatal Opioid Exposure on the Human Placental Methylome. *Biomedicines* **10**, (2022).
- Radhakrishna U, et al. Placental DNA methylation profiles in opioid-exposed pregnancies and associations with the neonatal opioid withdrawal syndrome. *Genomics* **113**, 1127-1135 (2021).
- Watanabe K, Taskesen E, van Bochoven A, Posthuma D. Functional mapping and annotation of genetic associations with FUMA. *Nat Commun* **8**, 1826 (2017).
- Deak JD, et al. Genome-wide association study in individuals of European and African ancestry and multi-trait analysis of opioid use disorder identifies 19 independent genome-wide significant risk loci. *Mol Psychiatry*, (2022).
- Shabalín AA. Matrix eQTL: ultra fast eQTL analysis via large matrix operations. *Bioinformatics* **28**, 1353-1358 (2012).
- Freshour SL, et al. Integration of the Drug-Gene Interaction Database (DGIdb 4.0) with open crowdsource efforts. *Nucleic Acids Res* **49**, D1144-D1151 (2021).
- Browne CJ, Futamura R, Minier-Toribio A, Hicks EM, Ramakrishnan A, Martinez-Rivera F, Estill M, Godino A, Parise EM, Torres-Berrios A, Cunningham AM, Hamilton PJ, Walker DM, Huckins LM, Hurd YL, Shen L, Nestler EJ. Transcriptional signatures of heroin intake and seeking throughout the brain reward system. *bioRxiv* 2023.01.11.523688.

REVIEWERS' COMMENTS

Reviewer #1 (Remarks to the Author):

Rompala et al have been largely thoroughly responsive to the reviews of all previous reviewers. This remains an important publication due largely to its novelty in examining hydroxymethylation in OUD post-mortem brain tissue. In spite of the responsiveness, several concerns remain.

Major: Given the heterogeneity of these samples (eg, all cases have PTSD, some controls had positive tox screens for opioids, it is very important that these limitations or weaknesses be noted/stressed in the discussion.

Minor:

The figures are not numbered.

The rightmost column in Table 1 reads "P-value" but two values are presented. Please include the additional information.

Figure 3 is not especially useful. It is not clear what information it is attempting to convey. What do the colors represent? In addition, it is not labeled to distinguish 5mC from 5hmC.

Figure 4 (A and B) should include column labels indicating that the values are correlations and p-values.

As requested, the authors added more information on the GWAS overlap, and an additional GWAS of OUD in that analysis, but the initial comment was regarding what was actually tested. That the authors used Fisher's exact test is not in itself completely informative. That suggests the data were arranged into 2x2 tables. Tables of what? That was the intent of the query in the original review. There are multiple ways by which an enrichment test can be performed and it would be helpful if those details were presented.

Reviewer #2 (Remarks to the Author):

I found the authors were very responsive to all the previous reviewer comments. I have no further comments at this time, with the exception of the manuscript need a strong grammatical review as there are a number of verb tense agreement issues, as well as other grammatical issues, throughout the manuscript.

Reviewer #3 (Remarks to the Author):

The authors have address all raised concerns adequately. This is a much improved manuscript.

May 22nd, 2023

We appreciate the reviewer's positive comments on our manuscript, highlighting its many strengths, and valuable suggestions to improve it further. We have addressed their comments and revised the manuscript accordingly.

The following is a point-by-point summary of these changes as well as responses to reviewers' concerns.

REVIEWERS' COMMENTS

Reviewer #1 (Remarks to the Author):

Rompala et al have been largely thoroughly responsive to the reviews of all previous reviewers. This remains an important publication due largely to its novelty in examining hydroxymethylation in OUD post-mortem brain tissue. In spite of the responsiveness, several concerns remain.

Major: Given the heterogeneity of these samples (eg, all cases have PTSD, some controls had positive tox screens for opioids, it is very important that these limitations or weaknesses be noted/stressed in the discussion.

We thank the reviewer for the suggestion. We have updated the limitation paragraph at the end of the Discussion to address this comment.

“The study is limited by the small sample size, but it is comparable to similar work in the human postmortem brain. Notably, our cohort is heterogeneous in terms of comorbidities and drug intoxication at the time of death. For example, all individuals with OUD were comorbid with PTSD. We addressed this issue by adding PTSD as a covariate in the model (PTSD is present in 50% of the non-OUD group) and replicated previous work assessing 5mC in the OFC of

individuals with heroin abuse¹². Furthermore, some of the samples from the non-ODU group were positive for opioid intoxication at the time of death. RNA sequencing data were conducted in bulk tissue, which may impede our ability to identify OUD DEGs and directly evaluate the impact of differential 5mC/5hmC on gene expression in a cell-type-specific manner. Another limitation is the sole inclusion of males, mostly of European ancestry. Future work will expand to female subjects and other ancestries to identify potential sex- and population-specific effects. Further, additional comorbidities (e.g., MDD and other SUDs) should be also examined. This study evaluated 5mC and 5hmC at CpG sites; future studies should evaluate the role of 5mC and 5hmC in non-CpG sites in the context of OUD, which have been suggested to play a crucial role in the development of neuropsychiatric diseases⁵⁰. Given the enrichment of differential 5mC and 5hmC genes and modules in GWAS studies, future work should evaluate potential methylation/hydroxymethylation quantitative trait loci to assess whether differential 5mC/5hmC is influenced by genetic background. Although the enrichment of GWAS signals may be indicative of potential causal effects of the OUD-associated 5mC/5hmC marks, research in model organisms (e.g., animal models, human induced pluripotent stem cells, brain organoids) could help confirm whether the identified marks are the cause or consequence of OUD.”

Minor:

The figures are not numbered.

All the figures are now numbered.

The rightmost column in Table 1 reads “P-value” but two values are presented. Please include the additional information.

We thank the reviewer for noting this. For simplicity, we updated the table to only keep the P-values.

Table 1: Demographics Summary.

	ODU- (N=26)	ODU+ (N=12)	P-value
Age (μ +/- SD)	43.1 +/- 11.55	37.6 +/- 8.9	0.1518
PMI (μ +/- SD)	30.65 +/- 8.15	29.6 +/- 7.5	0.7200
Ancestry			
AA	5	4	0.4428
EA	19	8	

Cigarette Smoking	13	10	0.0770
Alcohol Dependence	5	3	0.6893
PTSD	13	12	0.0026
Major Depressive Disorder	5	8	0.0086
Polysubstance Abuse	4	7	0.0174
Removed Outliers	2	0	1.0000

Figure 3 is not especially useful. It is not clear what information it is attempting to convey. What do the colors represent? In addition, it is not labeled to distinguish 5mC from 5hmC.

Figure 3 shows the gene-drug interaction analysis. Specifically, we show an Alluvial plot depicting the drug interactions (opioid-related drugs) with the annotated genes with OUD-associated differential 5mC and 5hmC CpG sites. Band color represents each of the annotated genes with interactions with opioid-related drugs. Band width indicates the number of interactions with opioid-related drugs. Panel A shows 5mC and panel B shows 5hmC. As shown, annotated genes with OUD-associated differential 5hmC CpG sites have a higher number of described interactions with opioid-related drugs. We have updated the manuscript to add this important information.

“Figure 3. Drug interaction analysis of the annotated genes with OUD-associated differential 5mC and 5hmC CpG sites. The Alluvial plot shows the number of described opioid-related drug interactions (right axis) for each annotated gene (left axis) with A) OUD-associated differential 5mC CpG sites, and B) OUD-associated differential 5hmC CpG sites. Band color represents each of the annotated genes with interactions with opioid-related drugs. Band width indicates the number of interactions with opioid-related drugs. ”

Figure 4 (A and B) should include column labels indicating that the values are correlations and p-values.

Following the reviewer's suggestion we have included a column label indicating correlations and p-values.

As requested, the authors added more information on the GWAS overlap, and an additional GWAS of OUD in that analysis, but the initial comment was regarding what was actually tested. That the authors used Fisher’s exact test is not in itself completely informative. That suggests

the data were arranged into 2x2 tables. Tables of what? That was the intent of the query in the original review. There are multiple ways by which an enrichment test can be performed and it would be helpful if those details were presented.

We thank the reviewer for the comment. We used FUMA for the GWAS enrichment analysis, which utilizes a hypergeometric test to perform the enrichment. Fisher's exact test was used specifically for the GWAS enrichment of OUD, not included in FUMA. This method considers the overlap of genes as previously described by Goeman & Bühlmann (2007). Both were performed to calculate the probability of a proportion of overlapping genes occurring by chance. We also considered the same background for both analyses, which is similar to the Metascape platform (30,182 genes).

Jelle J. Goeman , Peter Bühlmann, Analyzing gene expression data in terms of gene sets: methodological issues, *Bioinformatics*, Volume 23, Issue 8, April 2007, Pages 980–987, <https://doi.org/10.1093/bioinformatics/btm051>

"Genome-wide association studies (GWAS) enrichment analysis was conducted to test the probability of overlapping genes between the methylation analysis and published GWAS findings by chance. The analysis allows us to prioritize genes with a higher level of confidence, which could suggest a functional effect. The analysis was performed using the online website FUMA⁷⁰, which applies a hypergeometric test to perform the enrichment. First, the Ensembl IDs were transformed to Entrez IDs using David⁷¹ and submitted to FUMA's GENE2Function web tool. Once the enrichment analysis was generated, the results were merged with the domains and traits reported by the GWAS Atlas using in-house scripts. We further conducted the GWAS enrichment specifically for OUD using data from a recently published GWAS in European and African Populations²⁷. For this, we used Fisher's exact test and a similar background than the Metascape platform (30,182 genes)⁷⁰. Significance was defined as $FDR < 0.05$ for both analyses."

Reviewer #2 (Remarks to the Author):

I found the authors were very responsive to all the previous reviewer comments. I have no further comments at this time, with the exception of the manuscript need a strong grammatical review as there are a number of verb tense agreement issues, as well as other grammatical issues, throughout the manuscript.

We thank the reviewer for the positive comments. We have proofread the entire manuscript.

Reviewer #3 (Remarks to the Author):

The authors have address all raised concerns adequately. This is a much improved manuscript.